# Addressing Vitamin B$_{12}$ deficiency through aeroponic fortification of a salad crop (*Pisum sativum*)
Bethany M. Eldridge[1,9], Sree Gowrinadh Javvadi[2,9], Natalia Perez-Moral [2], Jessie Sweetman[3], Luíza Lane de Barros Dantas [1], Shikha Saha[2], Deirdre A. Lynch [1], Thomas Hunt [2], Sophie E. Clough[4,5], Jemal Toussaint[6], Andy Worrall[6], Lillian R. Manzoni[6], Nigel Robinson [4], Keara A. Franklin[3], Cathrina H. Edwards [2], Jonathan Clarke[1], Jack Farmer [6], Martin Warren[2,7,8] & Antony N. Dodd [1] ✉

Plants do not produce Vitamin B$_{12}$, creating a nutrient insufficiency risk for those who do not consume animal-derived foods without supplementation. Furthermore, various diseases cause Vitamin B$_{12}$ deficiency. Here, we establish an approach for B$_{12}$ dietary supplementation that harnesses a horticultural technology to deliver the recommended daily allowance (RDA) of B$_{12}$ within a single portion of a salad crop (pea shoots). We demonstrate the effectiveness of the approach in a commercial and scalable growing environment, conducted an economic evaluation, find that it has versatility for growers, does not alter the product shelf-life, and that the B$_{12}$ persists during cold-chain storage. Furthermore, the RDA of B$_{12}$ is bioaccessible from this crop during simulated human digestion. Taken together, this provides a commercially-viable approach for dietary supplementation of B$_{12}$ intake, and a roadmap for the development and evaluation of fortification strategies.

Vitamin B$_{12}$ is the only essential vitamin that is absent from plants, and therefore represents a nutritional insufficiency risk as people adopt more sustainable diets that minimise or exclude the consumption of meat and dairy[1]. Humans use the nutrient as a cofactor for just two enzymes, methionine synthase and methylmalonyl-CoA mutase[2]. The very long half-life of B$_{12}$ means that the nutrient is required in only minute quantities daily[2–5], with the USA recommended daily allowance (RDA) for B$_{12}$ being 2.4 µg/day for adults aged 16–65. Despite this low requirement for the vitamin, an estimated 6% of the UK adult population is B$_{12}$ deficient (i.e. ≤150 pM in serum) and a further 44% of the UK population is in the marginal or insufficient range (150–258 pM)[6].

Clinical hallmarks of B$_{12}$ deficiency are anaemia and neurological dysfunction of varying severity[2–4,7,8], as the nutrient is implicated in metabolic roles associated with the formation of red blood cells, myelin sheet formation, and neurotransmitter synthesis. Apart from dietary insufficiency, B$_{12}$ deficiency can also result from autoimmune gastritis, often referred to as pernicious anaemia, genetic factors associated with absorption, transport and activation, infections and parasites, and recreational drug use.

B$_{12}$ is the most structurally complex nutrient that is made exclusively by specific prokaryotes, and is completely absent from multicellular plants.

While ruminants including cows absorb B$_{12}$ produced by prokaryotes within their digestive tracts[9], the human gut microbiome appears to supply little, if any, B$_{12}$ to its host. Therefore, humans mainly acquire their B$_{12}$ from animal-based foods in their diet[3,4,10]. The major dietary sources of B$_{12}$ include fish, meat, poultry, eggs, milk, and other dairy products. Shellfish and algae are also thought to contain sources of B$_{12}$, but it is unclear whether these products contain analogues of B$_{12}$ that are active to humans[11,12]. B$_{12}$ forms that are processed and/or used directly by human cells include methylcobalamin, adenosylcobalamin, hydroxycobalamin and cyanocobalamin[13]. The biochemical synthesis of B$_{12}$ requires around thirty enzyme-mediated steps, which negates any possibility of engineering the biosynthetic pathway into edible plants in the near future.

Despite the presence of B$_{12}$ in many dietary sources, it is important for those that are B$_{12}$ deficient or insufficient to supplement their nutrition with B$_{12}$. However, tablet-based B$_{12}$ supplements are often ingested without food, which might prevent optimal B$_{12}$ absorption because intrinsic factor (a glycoprotein that complexes with B$_{12}$ and is essential for its uptake) is secreted by gastric parietal cells in response to eating[14,15]. An alternative approach to B$_{12}$ tablets is fortified food products, which can be consumed as part of a nutritious diet that promotes optimal B$_{12}$ absorption. This includes

[1]John Innes Centre, Norwich Research Park, Norwich, UK. [2]Quadram Institute Bioscience, Norwich Research Park, Norwich, UK. [3]School of Biological Sciences, University of Bristol, Bristol, UK. [4]Department of Biosciences, Durham University, Durham, UK. [5]Chemical and Biological Engineering, University of British Columbia, Vancouver, Canada. [6]LettUs Grow, Bristol, UK. [7]University of East Anglia, Norwich Research Park, Norwich, UK. [8]School of Biosciences, University of Kent, Canterbury, UK. [9]These authors contributed equally: Bethany M. Eldridge, Sree Gowrinadh Javvadi. ✉e-mail: antony.dodd@jic.ac.uk

fortification of a variety of foods, such as breakfast cereals, milk, and wheat flour[2], with $B_{12}$ fortification of flour occurring in many countries[16]. An alternative approach considered here is the fortification of fresh salad products.

The consumption of young salad leaves has gained popularity[17,18], with these salads reported to have enhanced nutritional profiles and contributing to dietary diversification[19]. For example, the microgreens market was valued at $1276.0 million in 2019 and is estimated to reach $2049.3 million by 2028, registering a compound annual growth rate of 11.1% from 2021 to 2028 (not inflation-adjusted)[20]. Within this market, there is a desire from growers and supermarkets to produce more nutritious salad crops to add value and health-benefit differentiators to their products, whilst fulfilling an important social role. We reasoned that there is potential to fortify salad products with $B_{12}$ during cultivation, since $B_{12}$ can be absorbed by certain plants when present in the growth medium[21–26] (Supplementary Table 1). Furthermore, existing patents indicate that some $B_{12}$ can be absorbed by the seeds of various crops[27,28]. However, it is unclear whether approaches that involve supplementing plant growth media with $B_{12}$ leads to a salad product that contains the RDA of $B_{12}$, whether it can be absorbed by human digestion, and whether such approaches are economically viable or scalable. Here, we developed an approach to fortify pea shoots with $B_{12}$ within a commercial growing environment, exploiting the rapid root development that occurs in aeroponic horticulture[29] to deliver a bioaccessible RDA of $B_{12}$ within a single salad portion.

## Results

### Aeroponic fortification of pea shoots with Vitamin $B_{12}$

We evaluated whether salads might be supplemented with $B_{12}$ during aeroponic horticulture in indoor farms, using pea shoots as an experimental model. We selected pea shoots (*Pisum sativum* CN Seeds cultivar 4019) because these represent a desirable crop for growers, as they have a rapid growth cycle, being ready for harvest within 10 days of germination in indoor horticultural environments, and this variety has tendril-free morphology that is better suited to industrial packing. Single or mixed pea shoot-containing salad bags are relatively popular with consumers because they add flavour compared to salads such as lettuce. A typical supermarket bag of pea shoots is 60–80 g, and we considered 15 g of pea shoots to be a single meal portion per adult when comparing fortification levels to the RDA.

We conducted this work in a vertical farm, using a commercially available aeroponics growing platform (Fig. 1a, b). Although such environments have greater environmental variability than laboratory-based experimental plant growth chambers, we selected this approach to evaluate the scalability of our approach. During cultivation, the aeroponically-supplied nutrient solution was supplemented with a form of $B_{12}$, cyanocobalamin, at various growth stages (Fig. 1a, b). We chose to use cyanocobalamin rather than other forms of the nutrient, because this is the most widely available form that is also bioactive for humans and has greater stability than adenosylcobalamin and methylcobalamin, which

photodegrade in seconds[30]. Because $B_{12}$ can be absorbed by seeds that are soaked in $B_{12}$ prior to germination[27] and can accumulate within the cellular vacuoles of some plant species[21], we reasoned that aeroponically cultivated pea shoots might absorb $B_{12}$ through their roots and accumulate $B_{12}$ within harvestable plant shoots. Furthermore, the aeroponic cultivation method leads to substantial root growth (Fig. 1b) and can increase root hair formation[31], potentially maximising interaction between the fortified nutrient solution and plant roots.

We supplemented the aeroponic cultivation nutrient solution with a range of cyanocobalamin concentrations, cultivated the plants for 8 days, and then harvested material for $B_{12}$ quantification using LC-MS analysis. We chose LC-MS rather than other methods because this can distinguish between the cobalamin forms that have human nutritional benefit and those that do not. In parallel, we replicated an existing seed-soaking method[27] and measured the quantity of $B_{12}$ within harvestable tissue. Supplementation of the aeroponic bed with cyanocobalamin for 48 h after transfer of seedlings to the growing system led to the accumulation of cyanocobalamin within the aerial portion of the plants (Fig. 2b). The quantity of $B_{12}$ that eventually accumulated reflected the dosing concentration (Fig. 2b). Supplementation with 10 μM of cyanocobalamin during aeroponic cultivation led to the accumulation of approximately 0.17 μg $B_{12}$ per gram of pea shoot tissue (Fig. 2b). In comparison, in a parallel experiment where seeds were soaked in cyanocobalamin solution prior to germination and cultivation[27], around 0.02 μg $B_{12}$ accumulated per gram of pea shoot tissue when cyanocobalamin was supplied at 10 μM (Fig. 2b). Therefore, a considerably greater quantity of $B_{12}$ accumulated within aerial plant tissue after $B_{12}$ supplementation during aeroponic cultivation, compared with an existing seed-soaking patent[27]. Using these data, we calculated the quantity of pea shoots that need to be harvested to obtain plant tissue that contains the US adult RDA of $B_{12}$ (2.4 μg/day), based on the $B_{12}$ levels accumulated during cultivation. At the greatest cyanocobalamin concentration tested (10 μM in this experiment), approximately 18 g of pea shoots contained the US $B_{12}$ RDA (Fig. 2b), raising the possibility that consumption of a relatively small quantity of pea shoots cultivated using this approach might deliver the RDA of $B_{12}$. In contrast, it would be necessary to consume >1.2 kg of pea shoots to obtain the RDA of $B_{12}$ from plants generated using the seed-soaking method (using 10 μM $B_{12}$; Fig. 2b), which seems unrealistic.

To optimize our approach, we were interested to identify the pea shoot tissues in which the $B_{12}$ accumulated during aeroponic supplementation with cyanocobalamin. Therefore, we conducted an experiment in which pea shoots supplemented with a variety of cyanocobalamin concentrations were separated into the germinated seeds, stems, leaves and roots, and the $B_{12}$ content was measured in pooled tissue from each of these organs. The greatest $B_{12}$ quantities accumulated within the leaves and roots (Fig. 2c, d), little was detected in the stems, and, in one independent repeat (Fig. 2c), some $B_{12}$ was present in the remains of germinated seeds. Although the overall $B_{12}$ accumulation varied between experimental repeats (Fig. 1c, d), a consistent feature was that $B_{12}$ supplementation at 10 μM and 20 μM

**Fig. 1 | Approach for the fortification of pea shoots with Vitamin $B_{12}$. a** Schematic of the process of fortification with $B_{12}$ during aeroponic horticulture. **b** Appearance of growing bed that is supplemented with cyanocobalamin. Pea plant roots are protruding from underside of jute matting, and red colour in aeroponic bed is caused by presence of cyanocobalamin. Created in BioRender. Dodd, A. (2026) https://BioRender.com/scq4r3p.

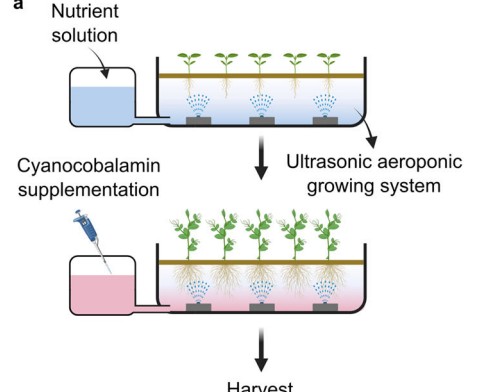
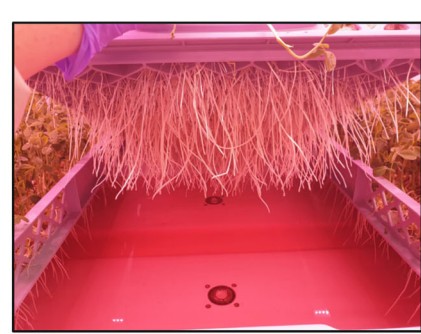

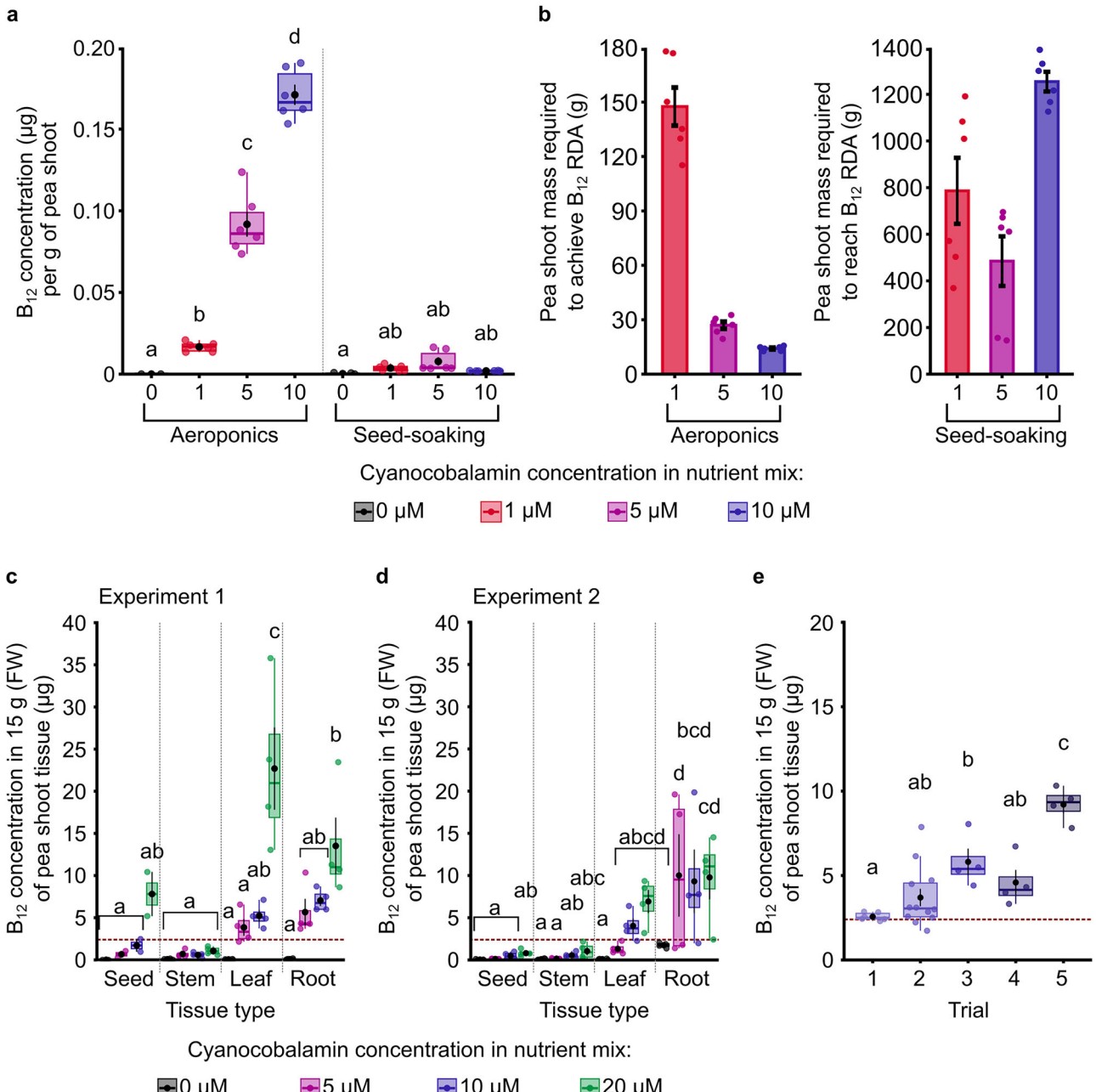

**Fig. 2 | Effective fortification of pea shoots with Vitamin B$_{12}$ during aeroponic cultivation. a** Comparison of the quantity of cyanocobalamin accumulated within pea plant leaves after cultivation with aeroponics, and an alternative method involving the soaking of seeds in cyanocobalamin during imbibition ($n = 6$; Statistical significance was determined using two-way ANOVA and Tukey HSD *post hoc*, with different letters indicating significant differences at $p \leq 0.05$). **b** Comparison of mass of pea shoots that must be harvested to obtain plant tissue that contains the USA RDA of B$_{12}$ (2.4 µg), when pea shoots were fortified using two different methods. Note the differing *y*-axis scales on the graphs for the two methods. **c, d** In two independent experiments, comparison of the quantity of cyanocobalamin present within several parts of the plants ($n = 4$, except for seed material in 1c, which has $n = 2$; statistical significance was determined using two-way ANOVA and Tukey HSD *post hoc*, with different letters indicating significant differences at $p \leq 0.05$). Pea shoots were harvested after 8 days of growth (3–4 node stage). **e** Across five independent trials, comparison of B$_{12}$ accumulation within pea shoot leaf tissue after 8 days of growth (3-4 node stage). Data are calculated on a per 15 g fresh weight basis (equivalent to one portion). Horizontal dashed line indicates the USA RDA of B$_{12}$ (2.4 µg). Statistical significance was determined using one-way ANOVA and Tukey HSD *post hoc*, with different letters indicating significant differences at $p \leq 0.05$; $n = 4–12$). In box plots, the box indicates the interquartile zone with the median line at the centre, whiskers indicate interquartile range, a black dot indicates the mean, and black lines connected to the dot indicate the s.e.m.

cyanocobalamin led to sufficient B$_{12}$ accumulation within the leaves to meet the RDA of B$_{12}$ in a portion of 15 g of leaf tissue. The roots also accumulated B$_{12}$ (Fig. 1c, d), but pea root tissue is unlikely to be attractive or palatable for consumers. This means that there would be some wastage of the B$_{12}$ molecule; if this represented an economic concern when growing at scale, future methods could be developed to liberate and reuse this B$_{12}$ from root tissue.

To examine variation in B$_{12}$ accumulation within leaf tissue, we conducted 5 independent growing trials, supplementing the plants with 10 µM cyanocobalamin, and compared B$_{12}$ accumulation in the leaves after 8 days of growth (Fig. 2e). The mean B$_{12}$ content always exceeded the USA RDA (2.4 µg) in 15 g of leaf tissue, although there was significant variation in B$_{12}$ levels between trials at the time of harvest (Fig. 2e). This variation could be due to factors such as gradients of environmental conditions, edge effects,

and planting density variation in the commercial growing environment. This is an important finding as it demonstrates that a cyanocobalamin dosing concentration at or exceeding 10 µM is sufficient to reach the $B_{12}$ RDA threshold irrespective of environmental variation across a commercial growing environment. Taken together, these experiments demonstrate that the leafy tissue of pea shoots supplemented with cyanocobalamin consistently accumulates the RDA of $B_{12}$ within a single 15 g salad portion.

## Flexibility of Vitamin $B_{12}$ fortification across pea shoot developmental stages

We determined that it is possible to achieve the RDA of $B_{12}$ within a relatively small quantity of pea shoots cultivated using aeroponics. We reasoned that it might be beneficial for growers to have flexibility in the developmental stage at which the plants are supplemented with $B_{12}$ during their cultivation. To investigate this, we cultivated pea shoots using the aeroponic platform and supplemented the growing media with cyanocobalamin for 48 h at three different developmental stages. The developmental stages at which cyanocobalamin supplementation occurred were shoot emergence, the 1–2 node stage and 3–4 node stage (Fig. 3a–c). Pea shoots supplemented with cyanocobalamin at the shoot emergence stage (Fig. 3a) accumulated less $B_{12}$ compared with those supplemented at the two later developmental stages (Fig. 3b, c). Although there was variation in node stage present at each sampling point due to some plant developmental heterogeneity across growing trays at a commercial sowing density, the greatest $B_{12}$ quantity always accumulated within the leaves (Fig. 3a–c). After 48 h of supplementation at all developmental stages, the leaves accumulated more than the quantity of $B_{12}$ that is necessary to deliver the RDA of vitamin $B_{12}$ within 15 g of plant material (Fig. 3a–c; dashed line across graph). This suggests that there is flexibility in the developmental stage at which the plants can be supplemented with cyanocobalamin to obtain a product that contains the RDA of $B_{12}$. This could benefit growers, who might wish to operate their horticultural facility in a flexible manner, according to staff availability and customer demand.

## Senescence and simulated shelf life are unaltered by Vitamin $B_{12}$ fortification

Senescence is a nutrient recycling process that reallocates resources within plants and can be initiated by age or stress. This causes leaf yellowing, which can reduce the shelf-life of harvested crops. We wished to determine whether our approach for the fortification of pea shoots with $B_{12}$ affects the senescence rate of the harvested pea shoots. This is important to evaluate whether the presence of $B_{12}$ might affect the shelf life of $B_{12}$-fortified salad products, relative to products that are not fortified with $B_{12}$. To investigate this, we used protocols that have been used previously to study the senescence of Arabidopsis leaves[32,33]. Pea shoots were cultivated as above and supplemented with cyanocobalamin at two concentrations (10 µM or 20 µM) for 48 h starting on day 4 of cultivation. After harvest on day 8, plant material was kept under conditions that simulated cold chain storage (5–6 °C in darkness) for up to 30 days (Fig. 4a). During this period, several established proxies for senescence[32] were obtained at regular intervals. These were the proportion of electrolytes leaking from plant tissue (a measure of cell lysis, which releases electrolytes), the chlorophyll content, and a measure of maximum photosynthetic efficiency[32]. Photosynthetic efficiency was estimated using the $F_v/F_m$ parameter derived from chlorophyll fluorescence analysis, which provides a measure of the maximum potential quantum yield of Photosystem II whereby a lower value indicates a lower potential quantum yield. Because the photosynthetic apparatus is dismantled during senescence-induced nutrient recycling, the chlorophyll content and $F_v/F_m$ can decrease during leaf senescence[32,34,35].

First, we monitored the $B_{12}$ content of harvested leaf tissue that was stored under simulated cold chain conditions (Fig. 4b), to evaluate whether the $B_{12}$ remained in the tissue during simulated storage. Batches of tissue were sampled at several intervals during cold storage, and $B_{12}$ content in leaves was measured using LC-MS. During this period of simulated cold chain storage, there was no significant alteration in the concentration of the

biologically active form of $B_{12}$ at either $B_{12}$ dosing concentration tested (Fig. 4b). This indicates that a portion of pea shoots fortified with $B_{12}$ using our approach still contained the RDA of $B_{12}$ after 4 weeks of simulated cold chain storage.

During dark storage of the detached leaf tissue, relative to the start of the time series, electrolyte leakage increased (Fig. 4c), chlorophyll content decreased (Fig. 4d), and $F_v/F_m$ decreased (Fig. 4e). This is consistent with the initiation of senescence in this leaf tissue. We did not detect accelerated leaf tissue senescence following $B_{12}$ fortification of this pea shoot variety, compared to the unfortified control, using these proxies for senescence. Over 28 days of storage under cold, dark conditions, tissue electrolyte leakage increased in the control (unfortified) tissue on days 26 and 28 post harvest, relative to earlier time points (Fig. 4c). However, there was no consistent significant difference between plants supplied with 10 or 20 µM of cyanocobalamin, and the control, at any time point (Fig. 4c). In general, there was also no significant difference in mean abaxial chlorophyll content between the control and fortified tissue (Fig. 4d), as measured using a non-invasive optical method (Dualex). The decrease in chlorophyll content was retarded slightly at days 27 and 30 in pea shoots supplemented with 20 µM cyanocobalamin relative to the untreated control (Fig. 4d). Furthermore, a measure of the maximum photosynthetic efficiency of photosystem II ($F_v/F_m$) was generally unaltered between the control and fortified material, with the exception of timepoints 28–30 days post harvest, where the control tissue had significantly lower photosynthetic efficiency than tissue fortified with 10 µM (Day 27) or 20 µM (Day 30) cyanocobalamin (Fig. 4e, f). Although the chlorophyll content and $F_v/F_m$ measures could suggest slightly delayed senescence in $B_{12}$-fortified pea shoot tissue compared with the control, this did not occur consistently across the $B_{12}$ concentrations tested so any effect is small. There was no obvious visible difference between the fortified and control pea shoots during simulated cold chain storage, with both undergoing some visible yellowing as tissue aged (Fig. 4g). Taken together, these data indicate that fortification of this pea shoot variety with cyanocobalamin, using the method that we developed, does not accelerate tissue senescence so cold-chain shelf life is unlikely to be altered.

## Simulated digestion of $B_{12}$-fortified pea shoots releases bioavailable Vitamin $B_{12}$

Our aim was to develop a fortification strategy to deliver the RDA of $B_{12}$, for at least one adult, within a salad portion (15 g). We reasoned that it is important to determine whether the $B_{12}$ accumulated within the pea shoot tissue is accessible for absorption during human digestion. Therefore, we implemented the Infogest 2.0 protocol to simulate in vitro the digestion in the upper gut[36]. With Infogest, it is possible to determine the quantity of a nutrient that becomes accessible for absorption at each stage of human digestion in the upper gut. The method involves the sequential incubation of the simulated meal (in this case, pea shoot tissue) with simulated digestive fluids and enzymatic mixtures that reflect the conditions found at each stage of human oral, gastric and duodenal digestion, and the release of nutrients at each of these stages (Fig. 5a)[36].

At the start of the in vitro Infogest digestion analysis, the effect of chewing was simulated in two ways: by mashing or by lysing the $B_{12}$-fortified pea shoots. We compared the quantity of $B_{12}$ that became accessible for absorption at the end of the simulated gastrointestinal digestion and confirmed that there was no difference between the quantity of $B_{12}$ released using these two simulated chewing methods (Fig. 5b, c). Therefore, tissue mashing is sufficient to release $B_{12}$ from the tissue (Fig. 5b). A meal portion (15 g) of pea shoots that was supplemented with 20 µM cyanocobalamin during cultivation released more than the RDA of $B_{12}$ (2.4 µg) during simulated digestion (Fig. 5c).

We then compared the quantity of $B_{12}$ released at each digestive stage, using mashed pea shoots supplemented with 10 µM or 20 µM of cyanocobalamin. This determined that the amount of $B_{12}$ released was similar after the oral and gastric digestive phases and increased significantly when the digesta reached the duodenal phase (Fig. 5d). Pea shoots supplemented with 20 µM cyanocobalamin showed the greatest quantity of $B_{12}$ release

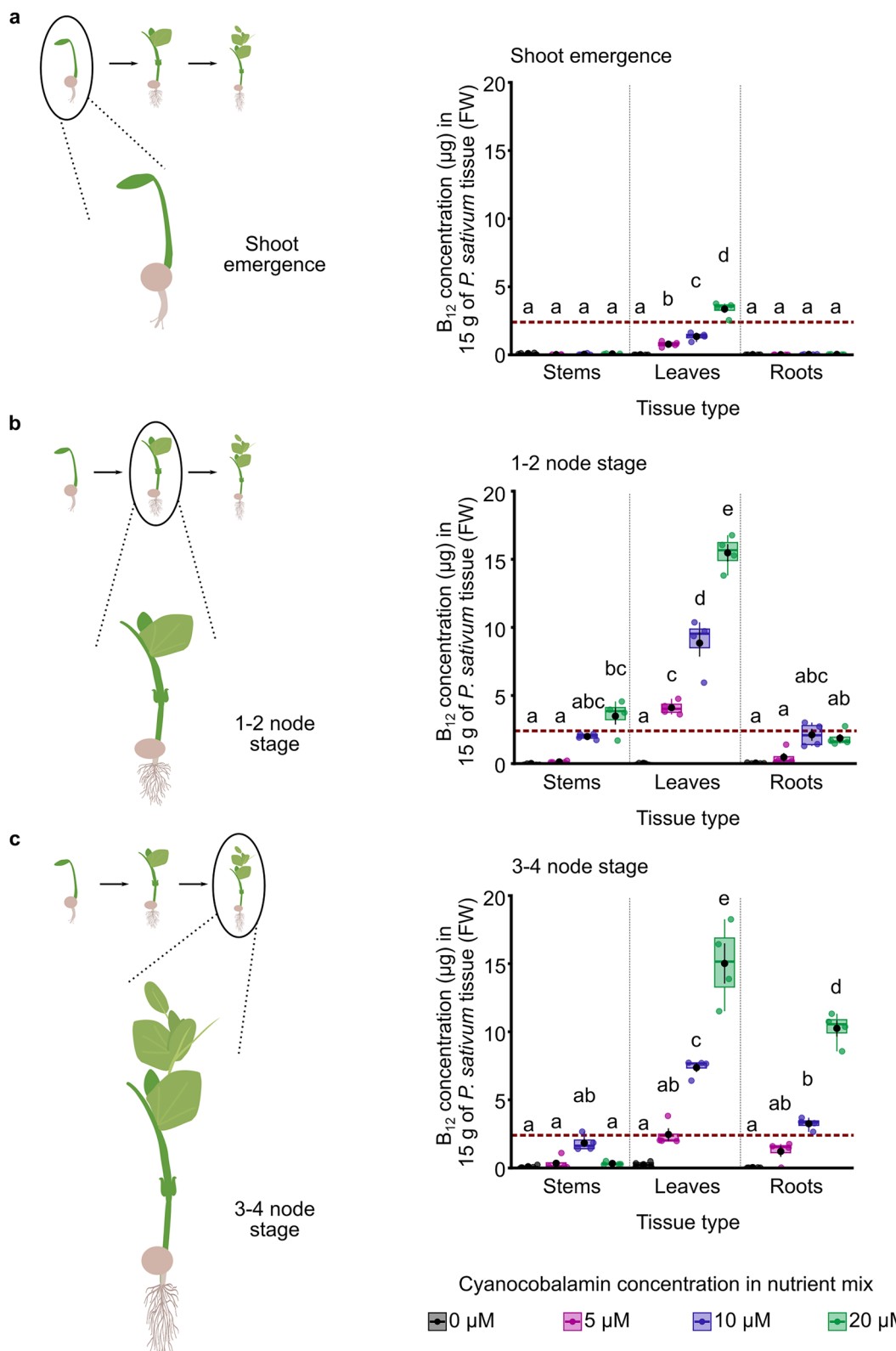

**Fig. 3 | Flexibility in developmental stage at which pea shoots accumulate the RDA of B$_{12}$.** Representative diagrams and B$_{12}$ content of 15 g (equivalent to one portion) of three parts of pea shoot plants, when supplemented with B$_{12}$ for 48 h at **a** shoot emergence stage, **b** 1–2 node stage and **c** 3–4 node stage (n = 4). Dashed horizontal line on graphs indicates 2.4 μg of B$_{12}$, which is the US RDA. Statistical significance was determined using two-way ANOVA and Tukey HSD post hoc analysis, with different letters indicating significant differences at $p \leq 0.05$). In box plots, the box indicates the interquartile zone with the median line at the centre, whiskers indicate interquartile range, a black dot indicates the mean, and black lines connected to the dot indicate the s.e.m. Parts created in BioRender. Dodd, A. (2026) https://BioRender.com/hhdlx3o.

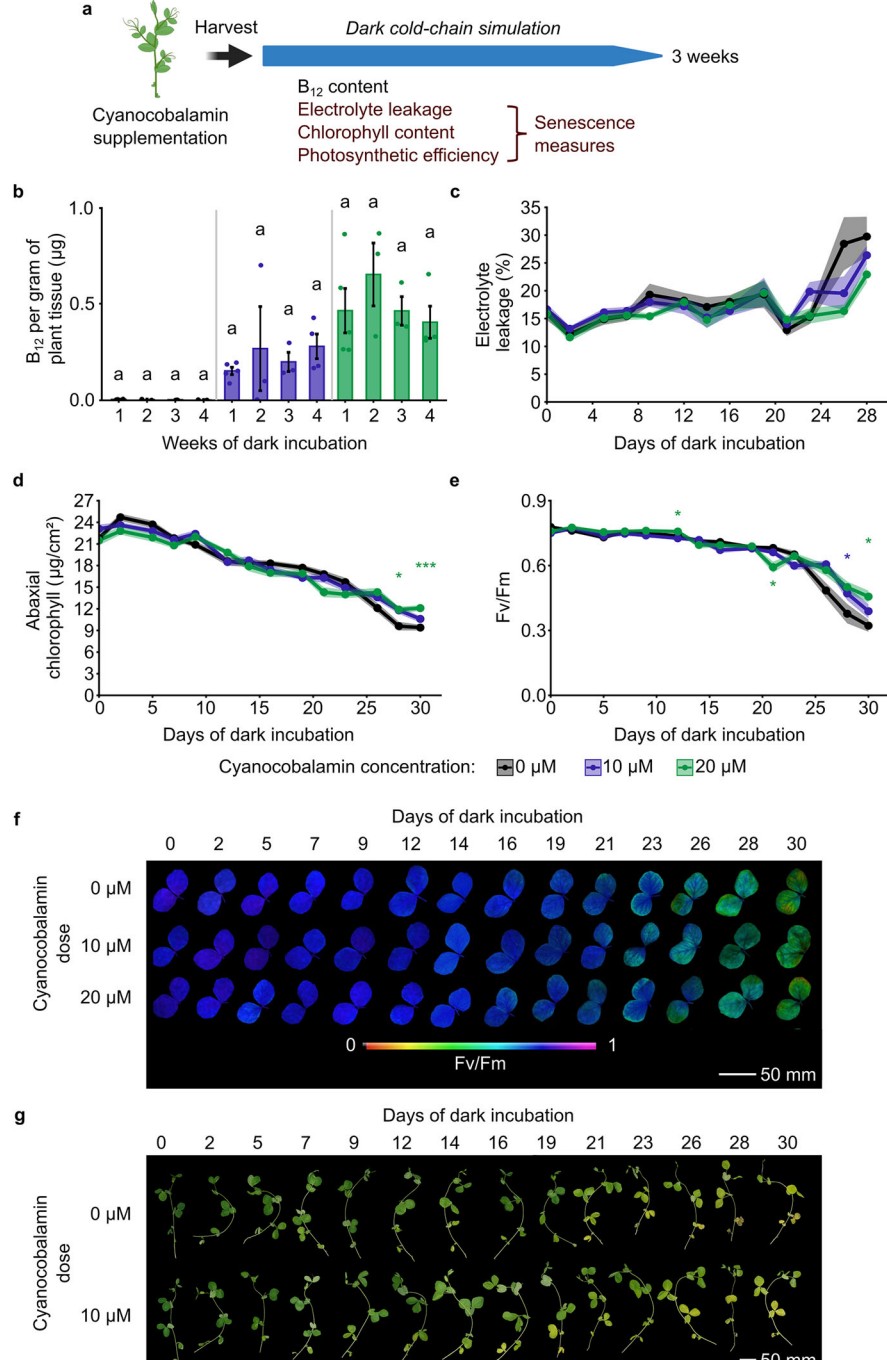

**Fig. 4 | Pea shoot fortification with B₁₂ does not accelerate senescence of harvested plant tissue during simulated cold-chain storage. a** Experimental workflow for investigation of post-harvest senescence after B₁₂ fortification. **b** During simulated cold-chain storage, comparison of B₁₂ content of harvested pea shoots previously supplemented with 10 μM and 20 μM during each week of storage ($n = 5$ (week 1), $n = 3$ (week 2), $n = 3$ (week 3), $n = 4$ (week 4), black bar = ± SEM. Statistical comparisons were performed within each B₁₂ treatment concentration, using one-way ANOVA and Tukey HSD *post hoc*, with different letters indicating significant differences at $p \le 0.05$). **c** Proportion of total electrolytes released from pea shoots supplemented with several concentrations of cyanocobalamin, compared with no treatment control, during 28 days of simulated cold chain storage. Measurements on successive days were from separate plants, to avoid resampling the same plant material ($n = 13$, shaded ribbons = ± SEM; no comparisons of the control against cyanocobalamin-treated plants were significant at $p \le 0.05$). **d** Mean abaxial chlorophyll content of pea shoots supplemented with several concentrations of cyanocobalamin, compared with no treatment control, during 28 days of simulated cold chain storage. Chlorophyll content was estimated using an optical method (Dualex

instrument) ($n = 14$, shaded ribbons = ± SEM; statistical comparisons are of cyanocobalamin-treated plants against untreated control). **e, f** Maximum efficiency of photosystem II (PSII) ($F_v/F_m$) of pea shoots, in plants supplemented with several concentrations of cyanocobalamin, compared with no treatment control, across 28 days of simulated cold chain storage ($n = 14$; shaded coloured ribbons = ± SEM; statistical comparisons are of cyanocobalamin-treated plants against untreated control). **f** Representative images of $F_v/F_m$ signal from pairs of pea shoot leaves during simulated cold chain storage. Images for each day of dark incubation are from different plants. **g** Visual appearance of representative pea shoots supplemented with 10 μM cyanocobalamin, with no treatment control, over 30 days of simulated cold chain storage. Images for each day are from different plants. Both treatments underwent a slight yellowing towards the end of the time-series. In **c–e**, asterisks indicate independent pairwise *t*-test comparisons of each treatment timepoint with the control (0 μM cyanocobalamin), where *$p < 0.05$; **$p < 0.01$; ***$p < 0.001$ (corrected for multiple testing using Bonferroni method); asterisks above or below lines indicate the B₁₂ concentration is significantly greater or lower than the control, respectively. **a** Created in BioRender. Dodd, A. (2026) https://BioRender.com/e463jgk.

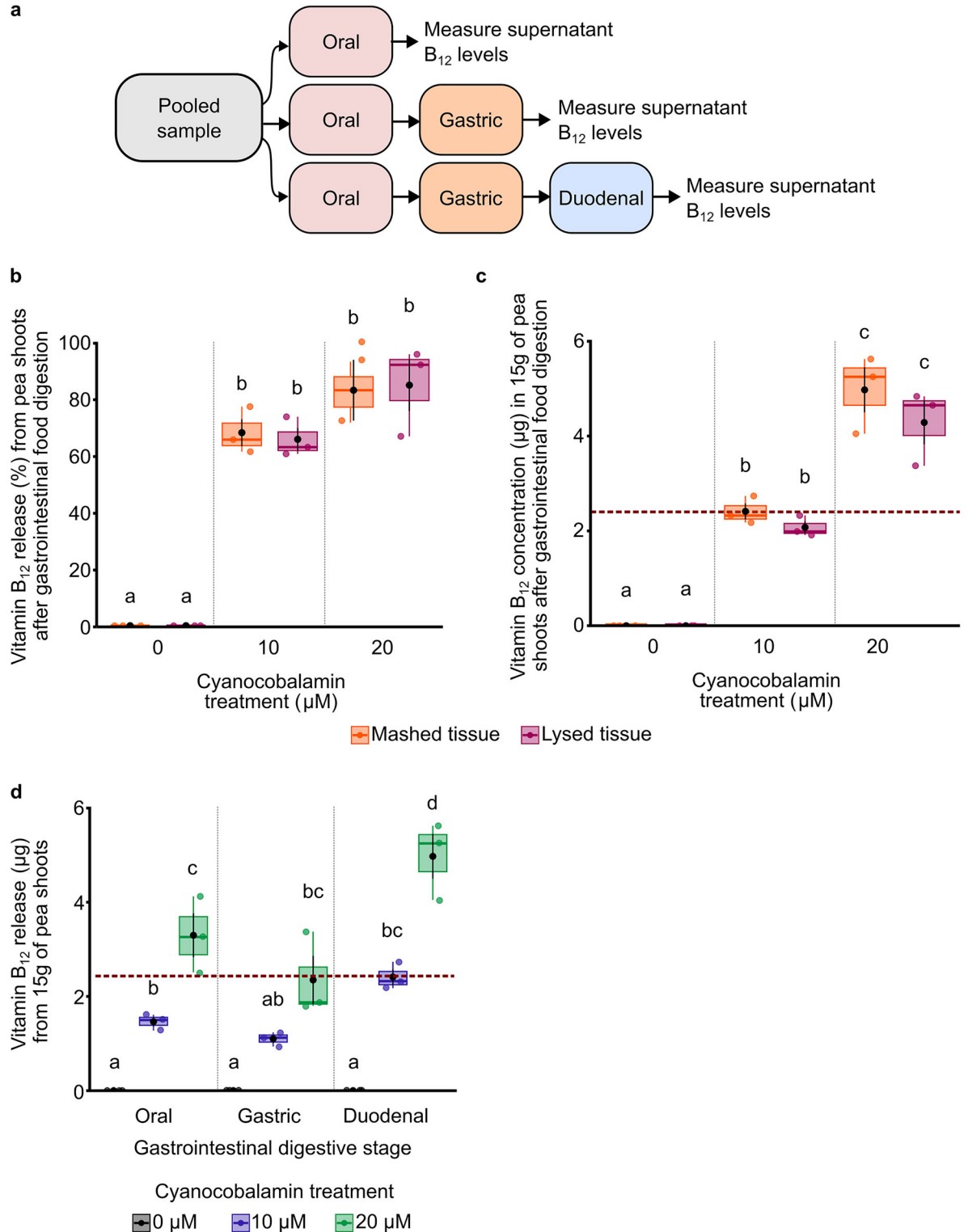

**Fig. 5 | INFOGEST analysis establishes that B$_{12}$ incorporated into pea shoots becomes accessible for absorption during simulated digestion. a** General INFOGEST workflow, showing steps involved in the measurement of bioaccessible B$_{12}$ at the end of the simulated oral, gastric and duodenal digestive phases. All assays were combined with enzymatic controls to eliminate any interference by the simulated digestive fluids or enzymes upon the measurement of vitamin B$_{12}$. **b** Proportion of B$_{12}$ released by simulated gastrointestinal digestion of pea shoots supplemented with two concentrations of cyanocobalamin. Pea shoots were either lysed (as for previous experiments) or mashed, to simulate chewing. **c** Quantity of B$_{12}$ released by simulated gastrointestinal digestion of pea shoots that was either mashed or lysed. Data are presented according to the quantity of B$_{12}$ that would be released from a 15 g portion of pea shoots, showing the 2.4 µg RDA for B$_{12}$ as a horizontal dashed line. **d** After supplementation with two concentrations of cyanocobalamin, the quantity of B$_{12}$ released from mashed pea shoot tissue within three digestive phases. Data are presented according to the quantity of B$_{12}$ that would be released from a 15 g portion of pea shoots, indicating the 2.4 µg RDA for B$_{12}$ as a horizontal dashed line. **b–d** $n = 3$; in box plots, the box indicates the interquartile zone with the median line at the centre, whiskers indicate interquartile range, a black dot indicates the mean, and black lines connected to the dot indicate the s.e.m. Statistical significance was determined using two-way ANOVA and Tukey HSD *post hoc* test, and significant differences at $p \leq 0.05$ are indicated with different letters.

during the duodenal phase. Significantly more $B_{12}$ was released from pea shoots supplemented with 20 μM cyanocobalamin compared with 10 μM (Fig. 5d; duodenal phase). Overall, these experiments identified that $B_{12}$ was released during simulated digestion of pea shoots fortified with $B_{12}$, using the strategy that we developed, and this $B_{12}$ would become available for digestive absorption.

## Economic analysis

We conducted a simple techno-economic analysis (TEA) to examine how variation in cyanocobalamin sourcing costs and fortified nutrient solution reuse influences commercial feasibility in an aeroponic farm. The economics described here reflect those at the time of writing (August 2025). $B_{12}$ is traded internationally and produced predominantly in China, which accounts for the majority of global supply[37–39]. Our TEA was not designed to model the full cost of pea shoot production, but instead to isolate additional costs arising from $B_{12}$ fortification using our aeroponic approach. Baseline production costs relevant to any commercially-produced pea shoots, such as seed, land, capital equipment and operational labour were not modelled, since they vary widely by growing system and region.

The TEA considered three input categories: the chemical cost of cyanocobalamin across sourcing tiers, specific energy costs of atomisers used for aeroponic mist generation, and labour specifically attributable to dosing the nutrient reservoir with cyanocobalamin. Labour and atomiser running costs were considered fixed, and three different price tiers for cyanocobalamin were used: wholesale food-grade, an active pharmaceutical ingredient (API) benchmark, and a pharmaceutical-grade laboratory-reagent. Cyanocobalamin fortification concentrations of 10 μM and 20 μM were evaluated, because our data show these concentrations are sufficient to fortify a single portion of pea shoots (15 g) to the RDA.

Our baseline scenario assumes no reuse of cyanocobalamin-supplemented nutrient solution. With this, the additional cost of fortifying 1 kg of pea shoots to the $B_{12}$ RDA using 10 μM cyanocobalamin was ~ £1.35/kg (wholesale), £1.56/kg (API), and £89–90/kg (laboratory reagent); and for 20 μM cyanocobalamin, ~£1.44/kg (wholesale), £1.67/kg (API), and £178–180/kg (laboratory reagent). On a per-portion basis (15 g pea shoots), this corresponds to an additional cost of ~£0.02–£0.023 (10 μM cyanocobalamin) and ~£0.022–£0.025 at (20 μM cyanocobalamin) when sourced from bulk or API supply chains, which we assume would be used commercially (Supplementary Fig. 2).

Because hydroponic/aeroponic systems often recirculate the nutrient solution, we also modelled the cost reduction through nutrient solution reuse, whilst estimating cyanocobalamin loss scenarios due to plant uptake and environmentally-induced degradation. Although direct measurements of cyanocobalamin depletion in recirculating systems are not available, studies of hydroponic operation have shown that nutrient recycling leads to nutrient depletion[40,41]. Cyanocobalamin is also sensitive to light-induced degradation in aqueous solution[30]. To reflect these combined risks, we applied progressive accumulative losses with reuse cycles in our model of 5% loss for two reuse cycles up to 25% loss for 10 reuse cycles. Despite estimated cyanocobalamin degradation, reuse markedly reduced fortification costs. After 10 reuse cycles, incremental costs for wholesale-sourced $B_{12}$ fell to ~ £0.61/kg at 10 μM cyanocobalamin and ~£0.66/kg at 20 μM cyanocobalamin, equivalent to ~£0.009–0.010 per 15 g portion, representing a ~55% cost reduction compared with no nutrient solution reuse. Similarly, for API-sourced $B_{12}$, costs reduced by 57% to ~£0.67/kg (10 μM) and ~£0.72/kg (20 μM), or ~£0.010 - £0.011 per 15 g portion. Therefore, while cyanocobalamin price dominates the economics, solution recycling could deliver large proportional savings. In the commercially relevant wholesale and API cyanocobalamin cost scenarios, recycling could reduce costs to below £0.01 per portion. These savings would scale significantly when applied to commercial production volumes.

## Discussion

Here, we developed an approach leading to the fortification of a single portion of pea shoots with the RDA of Vitamin $B_{12}$. The approach functions

effectively within a scalable commercial growing environment. We envisage that this could be used to produce a component for addition to packaged salads, to increase the options by which people can supplement their diet with $B_{12}$. Because the method allows the delivery of $B_{12}$ within a meal, rather than a tablet, we reason that its uptake during digestion will be optimised because food consumption will lead to favourable conditions for factors (haptocorrin, intrinsic factor) required for $B_{12}$ absorption. The approach does not alter the rate of senescence of harvested plant tissue, so seems unlikely to alter product shelf life. Given that the $B_{12}$ persists within the plant tissue for at least 4 weeks (Fig. 4b), the material could- in theory- be distributed to most locations globally if the economic case existed.

While we selected pea shoots for this work, it is possible that the approach could be adapted to other salad crops to increase its versatility. Previous studies reported that hydroponics can be used to fortify certain additional species with $B_{12}$ (Supplementary Table 1), suggesting this could be adapted to aeroponics. However, approaches with additional species require work to determine whether the $B_{12}$ is bioaccessible during digestion, is accumulated in forms that are active in humans, and to evaluate commercially viability. Through a direct comparison, we found that the approach we developed was several orders of magnitude more effective than a method described by a patent[27] that involves soaking pea seeds in $B_{12}$ solution (Fig. 2b). This is important because commercially available $B_{12}$ is expensive, so a $B_{12}$ accumulation level within plant tissue that meets commercial objectives is crucial for deployment at scale. We also conducted a small-scale comparison with hydroponic cultivation, which indicated that this can also be used to fortify pea shoots with $B_{12}$ (Supplementary Table 1), albeit requiring a larger nutrient tank volume and hence greater $B_{12}$ input cost. It is possible that aeroponic cultivation could be used to deliver other vitamins or minerals into salads in a relatively straightforward manner, although it would require experimental optimization and validation for any combination of nutrient and crop. Furthermore, we reason that the method could be combined with other fortification methods, such genetic approaches to increase iron accumulation within plant tissue, to fortify salads in a combinatorial manner. An alternative approach for salad fortification with $B_{12}$ has been proposed that involves co-culture and colonisation of plants with bacteria that over-produce $B_{12}$[42]. An advantage of this approach is that it eliminates the need to source and purchase food-grade cyanocobalamin, whereas the microorganisms may present difficulties with food safety regulators. This approach could combine well with indoor farming, although challenges with bacterial overgrowth, biofilm formation in the irrigation systems, and sterilization of the nutrient solution would need to be overcome.

Although our experiments used a legume, root nodules did not form because the relatively clean growing environment means that nod factor was likely absent (Fig. 1b), and the plants were grown for a short duration. However, the presence of microorganisms in certain soil types might present an alternative approach to fortify crops with $B_{12}$. For example, $B_{12}$ content of spinach was elevated when cultivated on organic soils containing farm animal manure[22], presumably due to microbially derived $B_{12}$. Other soil bacteria such as strains of *Pseudomonas fluorescens* can produce $B_{12}$[43], and the production of $B_{12}$ by rhizobia including *Sinorhizobium meliloti* opens future possibilities for legume fortification by $B_{12}$ derived directly from naturally associated bacteria[44]. As the strategy that we describe uses indoor farming technology that has increased energy requirements and requires capital expenditure, it might not be suitable for $B_{12}$ fortification in all regions globally, so approaches such as fortification using $B_{12}$-producing bacteria[42] could offer a less resource-intensive alternative. A further way that plant-based foods might be supplemented with $B_{12}$ is through fermentation, although it is unclear whether the $B_{12}$ present in foods such as natto contributes to serum $B_{12}$ levels[45].

Use of aeroponic cultivation could, in principle, improve $B_{12}$ uptake into the plant perhaps by allowing greater root respiration, or by altering root development or size[29] such that greater $B_{12}$ quantities can enter the plant. Importantly, aeroponic cultivation allows the use of a smaller volume of nutrient solution compared with hydroponics, thus reducing the overall

quantity of expensive $B_{12}$ that must be added to the nutrient mix. Specifically, the reduced volume of $B_{12}$ solution that is needed during aeroponic cultivation, combined with aerosolization, increases the probability that each root interacts with $B_{12}$ (compared with a larger volume of solution that would be required by hydroponics). Growers might wish to recapture the $B_{12}$ solution and use it again after root exposure to $B_{12}$. One challenge with recapture and reuse that will need to be overcome is to avoid cyanocobalamin degradation by the UV light sources used to sterilize the nutrient mix in indoor growing environments.

The mechanism(s) of $B_{12}$ uptake by the pea shoots are unknown and could be symplastic or apoplastic. Accumulation of $B_{12}$ in the leaves could occur through the action of the transpiration stream, given that cyanocobalamin is water-soluble. If this is the case, it might be possible to increase $B_{12}$ accumulation in the leaves through environmental treatments that maximise stomatal aperture during the $B_{12}$ treatment period, or that take advantage of the times of day of maximum stomatal aperture. The accumulation of $B_{12}$ in the roots could occur due to a restriction of uptake by the Casparian strip, but understanding this will require detailed knowledge of the uptake pathway. Greater $B_{12}$ accumulation after the cotyledon stage (Fig. 3) could simply reflect a greater tissue volume to store $B_{12}$ in larger plants, or developmental stage-specific changes in vascular configuration or root anatomy. Although identification of the uptake mechanism might allow for breeding of crop varieties that have greater efficiency of $B_{12}$ accumulation, this does not appear necessary in the case of the pea shoot variety that we tested because it was straightforward to obtain the RDA of $B_{12}$ within the leaf tissue of an existing commercial pea shoot variety. In the future, it will be useful to obtain more information about the uptake mechanism and location of $B_{12}$ storage within the plant tissue, since this could be important for extending the number of crop species to which the approach can be applied. It will also be informative to understand how different horticultural nutrient formulations affect $B_{12}$ uptake. A previous study using hydroponics and *Glycine max* (soybean) found that $B_{12}$ uptake was not impacted by a respiration inhibitor[24], suggesting that it is not an active process in soybean, but it is unknown whether this generalizes to other crops and growing conditions. If uptake is an active process that involves membrane transport mechanisms, we speculate that it might be possible to engineer active $B_{12}$ uptake from the growing media.

The approach that we developed used a commercial indoor farming environment. This overcomes several limitations that would arise from attempting $B_{12}$ fortification in outdoor environments. First, $B_{12}$ is degraded by the UV wavelengths present in sunlight[46], whereas indoor growing environments use either LED light sources that generally lack UV wavelengths or cultivation occurs under glass that filters UV from sunlight. Second, $B_{12}$ is degraded by various microorganisms (including soil bacteria)[47,48], so using a relatively clean indoor growing environment reduces the likelihood of $B_{12}$ loss by microbial activity. The method could be adapted for greenhouse cultivation, particularly when combined with aeroponic irrigation, but might need optimization of the light, temperature and humidity conditions that lead to the greatest efficiency of $B_{12}$ accumulation within the harvested tissue.

Our TEA suggests that aeroponic fortification pea shoots with $B_{12}$ is commercially viable, with minimal cost implications when wholesale or API supply chains are used for cyanocobalamin. Importantly, this TEA did not attempt to capture the full cost structure of fortified pea shoot production—which varies widely with many operational costs such as seeds, infrastructure, and operational labour. Instead, we wished to estimate the additional cost introduced by $B_{12}$-fortification under our aeroponic dosing method. Our analysis also predicts that nutrient solution reuse will increase cost efficiency. Cyanocobalamin degradation will depend on local environmental conditions, so cyanocobalamin loss coefficients during reuse may require customization to individual commercial growing environments.

## Conclusion

We have developed a straightforward and commercially viable approach for the fortification of a salad crop with a quantity of Vitamin $B_{12}$ that exceeds

the RDA. Given that indoor farms are amenable to interventions that raise the nutritional value of horticultural products[49], $B_{12}$ fortification could be stacked with other types of fortification to further enhance the nutritional content and commercial value of the product. Focusing on multiple aspects of $B_{12}$ fortification in a single species has enabled us providing a roadmap for this type of crop fortification in other species and with other nutrients, from the underpinning biology to an economic evaluation. The approach described here represents one strategy for combating $B_{12}$ insufficiency or deficiency, and has the capacity for commercial deployment for production at scale.

## Methods

### Plant material and growth conditions

We selected a commercial variety of *Pisum sativum* (CN Seeds cultivar 4019) for this work. Cultivar 4019 is suitable for indoor farm production because it grows rapidly and lacks tendrils, providing greater compatibility with automated processing and packing equipment. In all cases, the pea shoots were cultivated in an experimental aeroponic indoor farm (LettUs Grow, Bristol, UK) that uses commercially available equipment (Supplementary Fig. 3a). The cultivation system used an aero-hydro cultivation method[29], which has undermounted ultrasonic devices that produce a nutrient mist at regular intervals from the surface of a thin layer of water at the base of the growing bench[50]. This produced an aerosol with particle sizes ranging 5–15 μm at the point of landing. In this vertical farming system, each cultivation stack comprised 4 layers of growing shelves and lighting (Supplementary Fig. 3b).

To induce germination, dried pea seeds were imbibed in cold water for 2 days with gentle agitation. The seeds were pre-rinsed to clean surface debris, and then transferred to the imbibition tank. During imbibition, agitation occurred through the action of a pump that supplied aerated water to the imbibition system for 15 min during each hour. This water was refreshed at least once per day. Imbibed peas were subsequently transferred to jute horticultural matting that was pre-saturated with water and placed on horticultural trays. Imbibed peas were sown at a density of 357 g per horticultural tray. These were transferred to a dark germination room that was maintained at 26 °C and 90% humidity, for 3 days. At this point, cotyledons had emerged, and radicles were sufficiently long to pass through the jute matting. After this, horticultural trays were transferred to the aeroponic cultivation environment (grow beds), with each aeroponic bed holding four trays. Aeroponic mist generation was operated with a duty cycle of 1 min misting on, 4 min misting off. Pea shoots were cultivated in the aeroponic system for 8 days to the 3–4 node stage, and for shorter durations for comparisons of developmental stages (Fig. 3). For all experiments, the growth medium (nutrient mix) was VitaLink HydroMax nutrient A+B mixes (Supplementary Table 2) (HydroGarden, Coventry, UK), maintained at pH 6.2 and an electrical conductivity of 1.7 μS cm$^{-1}$. Growing conditions comprised a 16 h photoperiod with 24 °C day temperature, 17 °C night temperature, and 70–75% relative humidity). Lighting was provided with Valoya AP673L horticultural LED sets (Valoya, Helsinki, Finland; plant height PAR of 150 μmoles photons m$^{-2}$ s$^{-1}$). The nutrient solution temperature was 17 °C. Chamber $CO_2$ levels were approximately 800 ppm. The cultivation bed volume was 20 L, and the reservoir volume was 1000 L. In addition to automated farm monitoring, during experimentation, the nutrient mix pH was monitored using a Hanna Instruments HI-98128 Pocket pHep5 Water-resistant pH tester (Hanna Instruments, Rhode Island, USA), and electrical conductivity and grow bed temperature measured using a Hanna Instruments HI-991300 electrical conductivity (Hanna Instruments, Rhode Island, USA), total dissolved solids, and a temperature meter.

### Aeroponic Vitamin $B_{12}$ supplementation methods

The $B_{12}$ analogue cyanocobalamin was used for all experiments. Cyanocobalamin (Merck, Darmstadt, Germany) was prepared as 1 mM stocks in $H_2O$ up to a week before each experiment. Cyanocobalamin stocks were stored at 4 °C in light-excluding conditions to ensure stability of the

molecule. Cyanocobalamin was added at concentrations of 10–40 µM, with the quantity of stock solution added calculated according to the total volume of nutrient solution in the growing system. Cyanocobalamin was added only once to the growing system; we did not monitor or top up its concentration, and its addition was conducted manually. Cyanocobalamin can degrade outside a pH range of 4–6.5 and in the presence of UV light[51] (one study indicates it has greatest stability at pH 6.3[52]). Therefore, for the duration of the cyanocobalamin treatment of the plants, the pH was carefully controlled (pH 6.2, as above), and the nutrient mix was disconnected from the reservoir so that it was not exposed to a sterilizing UV treatment. During experiments that replicated a patent[27] where $B_{12}$ was added during seed imbibition, the cyanocobalamin was added at the required concentration to the seed imbibition tank, prior to germination.

## Tissue sampling

For all experiments, sampling of pea shoot tissue started at dawn, following an aeroponic cultivation period of either 8 days (Figs. 2, 4 and 5) or shorter periods (Fig. 3). Tissue was harvested on a per-horticultural-tray basis, with pea shoots cut to the base of the jute matting using a single-cut method. When needed for $B_{12}$ measurements (Fig. 2), roots were cut from underneath the jute mat. The fresh weight of tissue was measured at the time of sampling, and in the case of roots, excess water was carefully squeezed from the roots before weighing (this was necessary because aeroponic irrigation leads to considerable amounts of water within the roots). For analytical purposes, tissue was obtained from five locations within each cultivation tray to obtain representative information about $B_{12}$ accumulation across the growing tray (Supplementary Fig. 3c). Tissue from the five locations on each tray was pooled and comprised one replicate for analytical purposes. A minimum of 4 replicate trays were sampled per treatment (specified in figure legends). Sampling was conducted by 3–4 people working as a team, with team members blind to the cyanocobalamin treatments. Tissue was placed into 50 ml Falcon tubes and frozen in dry ice prior to longer-term storage at −80 °C before analysis.

## Vitamin $B_{12}$ extraction and quantification

For quantification of $B_{12}$ content, a minimum of 50 mg of frozen tissue was lysed using a TissueRuptor II (Qiagen, Hilden, Germany). 5 ml of extraction buffer (50 mM HEPES, 200 mM NaCl, 1 mM DTT, 1% (v/v) Triton-X 100, pH 7.8) and an internal standard were added to the sample prior to vortexing for 5 min. The samples were incubated at 80 °C for 40 min to release the bound $B_{12}$, then centrifuged. The supernatant was removed and retained. The pellet was lysed again with a TissueRuptor II and a further 5 ml of extraction buffer added, before vortexing and centrifugation, to ensure all bound vitamin $B_{12}$ was released from the sample into the supernatant. The supernatants were pooled, and 1 ml of this was filtered on a spin column to remove any remaining debris from the sample for analysis.

20 µl of sample was injected for LC-MS analysis of $B_{12}$ content. The vitamin $B_{12}$ concentration was resolved on a UPLC HSS T3 1.8 µM column with 0.1% (v/v) formic acid in water, and 0.1% (v/v) formic acid in acetonitrile with a gradient flow for 20 min on an LC-MS instrument (Agilent 1290 infinity II; Agilent, California, USA). Through multiple reaction monitoring, the vitamin $B_{12}$-specific ion 678-147 was detected and quantified. The $B_{12}$ was quantified using matrix match calibration ranging (0–100 ng ml$^{-1}$) using a vitamin $B_{12}$-free control tissue sample.

## Senescence experiments

For all measures of senescence (electrolyte leakage, chlorophyll content, chlorophyll fluorescence; Fig. 4), pea shoots were cultivated for 8 days, to the 3–4 node stage. The aerial tissue was harvested as described previously, and stored in sealed bags, in darkness, at 4 °C, to simulate cold chain storage. During this period of storage senescence measures were obtained at regular intervals (Fig. 4b–g). Electrolyte leakage is a destructive measure, whereas measurements of chlorophyll content and chlorophyll fluorescence are not destructive.

## Electrolyte leakage

Electrolyte leakage due to cell degradation during senescence was measured using an established method[53]. For each timepoint, separate batches of tissue were used for analysis to avoid resampling the same leaves. For each replicate at each timepoint, three 5 mm discs were cut from separate leaves of an individual plant using a cork borer and placed in a glass test tube that was pre-washed with MilliQ water. 5 ml of MilliQ water was applied to the leaf discs, and the tubes were shaken at 120 rpm for 3 h at room temperature. The water was removed, and its electrical conductivity was measured ($EC_{pre}$). To release all electrolytes, leaf discs were transferred to a clean tube, and frozen to −70 °C for 1 h and then allowed to thaw gradually[53–55]. This ruptured the tissue and release all remaining cellular electrolytes. The 5 ml of water removed previously was returned to the tube, which was shaken at 120 rpm for 3 h at room temperature. Subsequently, the electrical conductivity of the water was measured again ($EC_{frz}$). The proportion of total electrolytes leaking from the tissue was expressed as [$EC_{pre}$ / $EC_{frz}$] * 100.

## Chlorophyll content

Chlorophyll levels were estimated using a non-destructive optical method (Dualex 4, Force-A Scientific, Paris, France). Briefly, the instrument was clamped onto replicate leaves and measurements obtained according to the manufacturer's instructions. The Dualex instrument measures the ratio of light transmitted through the leaf at 710 nm and 850 nm, and calculates the chlorophyll content according to the manufacturer's research[56]. For each replicate plant, the mean of 3 or 4 measurements was obtained from the abaxial surface of each leaf.

## Chlorophyll fluorescence

Because plants were small and fragile, we used chlorophyll fluorescence imaging rather than an approach using a leaf clip and fibre optic. The ratio of variable chlorophyll fluorescence to maximum chlorophyll fluorescence ($F_v/F_m$) was measured using an Imaging-PAM M series (MAXI Version, Walz, Effeltrich, Germany). Prior to use, the aperture and gain settings were adjusted to ensure that the maximum fluorescence signal was between 1.5 and 2 (arbitrary units), to provide an appropriate dynamic range. Leaves were dark-adapted for 20 min prior to measurement by placing the plant material in a light-excluding container, and the $F_0$ (minimum fluorescence level in the dark-adapted state) and $F_m$ (maximum fluorescence yield following a saturating light pulse) parameters were obtained. A saturating pulse of instrument intensity setting level 2 was used[57]. $F_v/F_m$ was calculated as ($F_m$-$F_0$)/$F_m$. Data were processed initially using ImagingWIN software (Walz), which was also used to control the imaging system. Each leaf was considered to be one individual replicate, with the mean fluorescence signal obtained across the entire leaf surface and all leaves on the plant measured.

## Comparison with hydroponics

*P. sativum* seeds were germinated and grown in compost at for 8 days, at room temperature. Seedlings were selected randomly for subsequent experimentation. After 8 days of growth, excess soil was removed from roots and plants were cultivated hydroponically for 2 days without cyanocobalamin supplementation. Roots were then submerged 10 µM cyanocobalamin for 48 h. Tissue above the second node was detached, frozen at −80 °C, and ground in liquid $N_2$ before storage at −80 °C prior to $B_{12}$ extraction and analysis[58,59].

## INFOGEST simulation of digestive $B_{12}$ release

For analysis of digestive $B_{12}$ bioaccessibility (Fig. 5), pea shoots were cultivated for 8 days, to the 3-4 node stage. Tissue was in subsequent analyses within 2 days of harvest. Vitamin $B_{12}$ bioaccessibility in pea shoot leaves was assessed by simulating their digestion in vitro in the upper gastro-intestinal tract, following the Infogest 2.0 static method[36]. Infogest has been used successfully to measure digestive accessibility of vitamin $B_{12}$[60,61]. Digestions were conducted in triplicate with a simultaneous control digestion run, within which water replaced the pea shoots to account for background signal arising from the added fluids, enzymes, and bile salts. Enzyme activities and

bile salt concentrations were calculated beforehand. Pepsin from the porcine gastric mucosa (Sigma P7012); pancreatin from porcine pancreas (Sigma P7545); bovine bile (Sigma B3883) and salts used in the preparation of the simulated digestion fluids (KCl, $KH_2PO_4$, $NaHCO_3$, NaCl, $MgCl_2(H_2O)_6$, $(NH_4)_2CO_3$, HCl and $CaCl_2(H_2O)_2$) were obtained from Sigma-Merck.

In brief, a meal consisting of 0.4 g of either smashed or lysed pea shoot leaves with 0.6 ml of water was mixed at 40 rpm using an orbital shaker, at 37 °C. To this mixture, 0.8 ml of simulated salivary fluid, 5 µl of 0.3 M $CaCl_2$, and 0.195 ml of water were added at pH 7 for the oral phase. After 2 min, the gastric phase was initiated by adding 1.6 ml of simulated gastric fluid, reducing the pH to 3 with 1 M HCl, followed by addition of 1 µl 0.3 M $CaCl_2$, 0.264 ml water, and 0.1 ml of a pepsin solution to achieve 2000 units of pepsin per ml in the final gastric digestion mixture. The duodenal phase commenced 1 h later by adding 1.7 ml of simulated intestinal fluid to the gastric chyme, increasing the pH to 7. Then, 8 µl of 0.3 M $CaCl_2$, 0.792 ml water, 0.5 ml of bile solution (yielding a final concentration of bile salts of 10 mM), and 1 ml of pancreatin (to achieve 100 units of trypsin per ml of final digestion mixture) were added to the digestion mixture. The duration of the duodenal phase was 3 h, after which the digested samples were boiled at 99 °C for 5 min to inactivate the enzymes. The digested mixture was then centrifuged at $4500 \times g$ for 10 min to separate the released nutrients in the supernatant from the pellet. Both supernatant and pellet were subsequently frozen for storage before being analysed.

## Techno-economic analysis

The TEA was designed to identify additional costs of $B_{12}$ fortification of pea shoots in aeroponic horticulture, rather than model the full economics of pea shoot production. This is because the latter will vary substantially by growing platform, geography, and local markets. We specifically considered the additional inputs that are required for our fortification approach: (1) chemical costs of cyanocobalamin across sourcing tiers, (2) energy use costs of aeroponic atomisers in our mist-based $B_{12}$-fortification method, (3) additional labour time for dosing with cyanocobalamin. Costs of pea shoot production such as seed, land, and baseline energy requirements of the production process were excluded, since they are growth system and region-specific. Our aim was to estimate the additional costs arising from $B_{12}$-fortification of a kg or portion of fortified pea shoots.

We modelled additional costs of pea shoot Vitamin $B_{12}$ fortification at 10 µM and 20 µM cyanocobalamin supplementation in an aeroponic system, in a glasshouse with natural lighting. In the model, growing beds each contained 20 L of nutrient solution and produced 4 kg of pea shoots per growth cycle, i.e. 5 L kg$^{-1}$ nutrient solution application volume. The energy cost for ultrasonic atomisers was fixed at £0.432/kg per cycle, and cyanocobalamin dosing labour was modelled at £0.05/kg based on a farm operator spending 1 min dosing $B_{12}$ into each aeroponic bed. Labour cost was benchmarked to the UK National Living Wage of £12.21/hr (UK Government, 2025).

Cyanocobalamin costs were parameterised using three sourcing tiers: wholesale food grade (£1.30/g; Made-in-China (aggregates supplier quotes from major Chinese manufacturers), accessed August 2025), API (£1.56/g; PharmaCompass, accessed August 2025), and laboratory grade reagent (£89.1/g; Thermo Fisher, accessed August 2025). Because wholesale and API prices for $B_{12}$ are not reported systematically in the scientific literature, we drew from representative commercial listing sources commonly used in TEA. API benchmarks were obtained from PharmaCompass, an industry platform that compiles manufacturer and trader data. Pharmaceutical-grade laboratory reagent prices were from the ThermoFisher Scientific catalogue. These sources were used to provide realistic input assumptions for cost modelling, rather than to serve as definitive price references.

Loss fractions for nutrient recycling were informed by evidence of micronutrient depletion in recirculating hydroponics[40,41] and stability studies of cyanocobalamin in aqueous systems[30]. To capture plausible degradation from microbial metabolism, precipitation, filter-induced light exposure, and solution handling, cumulative losses were conservatively assumed at 5%, 10%, 15%, 20%, and 25% at 2, 4, 6, 8, and 10 reuse cycles,

respectively. To achieve the target dosing concentrations in a 20 L aeroponic bed, supplementation from a 1 M cyanocobalamin stock would be 0.2 mL (10 µM) and 0.4 mL (20 µM). These values were used as a dilution check to confirm the consistency of cost calculations.

The following equations were used to estimate the additional cost of fortifying pea shoots with vitamin $B_{12}$ using dosing concentrations of 10 or 20 µM, with and without nutrient solution reuse. To calculate the cost of cyanocobalamin per litre of nutrient solution at a given dosing concentration of cyanocobalamin (Eq. 1):

$$C_l = (C_g \, M \, d) \times 10^{-6}$$

Where $C_l$ is the cost of cyanocobalamin per litre, $C_g$ is the per gram cost of cyanocobalamin, $M$ is the molecular weight of cyanocobalamin (1355.4 g mol$^{-1}$) and $d$ is the desired dosing concentration in µM (typically, 10 or 20 µM).

To calculate the cost of cyanocobalamin per kg of pea shoots produced, a simple multiplication was used because 5 L of nutrient solution are required to produce 1 kg of pea shoots (Equation 2):

$$C_{kg} = 5C_l$$

Where $C_{kg}$ is the cost of cyanocobalamin per kg of pea shoots, $C_l$ is the cost of cyanocobalamin per litre of nutrient solution used from Equation 1, and 5 L of nutrient solution are required to produce 1 kg of pea shoots.

When considering the reuse of nutrient solution, the reuse-adjusted cyanocobalamin cost per kg of pea shoots produced accounting for nutrient losses was (Equation 3):

$$C_{ra} = C_{kg} \left( \frac{N - (N-1)r}{N} \right) \frac{1}{1 - L_N}$$

Where $C_{ra}$ is the reuse adjusted cost of the nutrient solution, $C_{kg}$ is the cost of cyanocobalamin per kg of pea shoots (Equation 2), $N$ is the number of solution reuse cycles (e.g. 2, 4, 6, 8), $r$ is the fraction of the nutrient solution that is reused (1 in this model, representing complete reuse), and $L_N$ is the cumulative cyanocobalamin loss fraction after N reuse cycles (assumed as 0.05, 0.10, 0.15, 0.20 or 0.25, respectively).

The total additional cost of fortifying pea shoots with cyanocobalamin, per kg, was calculated as (Equation 4):

$$C_f = C_{ra} + 0.432 + 0.05$$

Where $C_f$ represents the total additional cost of fortification per kg, Cra is the reuse adjusted cost of cyanocobalamin within the nutrient solution (Equation 3), £0.432 represents the fixed energy cost of ultrasonic atomisation per kg of pea shoots, and £0.05 represents the estimated labour cost.

## Statistics and reproducibility

Details of replication levels and specific statistical analyses used are provided in the figure legends. The TEA models derive from simulations, thus lack statistical analyses. Sample sizes: Fig. 2a ($n = 6$), Fig. 2c, d ($n = 4$, except Fig. 2c where $n = 2$), Fig. 2e ($n = 4–12$), Fig. 3a–c ($n = 4$), Fig. 4b ($n = 5$ (week 1), $n = 3$ (week 2), $n = 3$ (week 3), $n = 4$ (week 4), Fig. 4c ($n = 13$), Fig. 4d ($n = 14$), Fig. 4e, f ($n = 14$), Fig. 5b–d ($n = 3$), Supplementary Fig. 1a–I ($n = 14$). All statistical analyses and data presentation used R v4.5.1 (https://www.r-project.org). R packages used for analysis and figure preparation were ggplot2, scales, reshape2, cowplot, gridExtra, ggthemes, ggpubr, tidyverse, dplyr, multcompView and rstatix. All underlying source data for main figures are in Supplementary Data 1.

## Reporting summary

Further information on research design is available in the Nature Portfolio Reporting Summary linked to this article.

## Data availability

Source data are included within the Supplementary Information of this study. Data for the graphs in the main figures are within Supplementary Data 1. All other data are available on reasonable request from the corresponding author.

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

## Acknowledgements

This work was funded by UKRI-BBSRC (Follow on Fund BB/X512229/1, FTMA BB/S507921/1, BB/W015749/1, SWBio DTP BB/M009122/1 and BB/T008741/1, NRPDTP BB/T008717/1, BBSRC Institute Strategic Programmes GEN BB/P013511/1, BRiC BB/X01102X/1, and FMH BB/X011054/1 with its constituent project BBS/E/F/000PR13630), the Wellcome Trust (EDESIA: Plants, Food and Health PhD programme 218467/Z/19/Z), and Innovate UK (for the Aeroponic Research & Development Centre at LettUs Grow). AND is funded by the European Union (ERC, MicroClock, 101166968). Views and opinions expressed are those of the author(s) only and do not necessarily reflect those of the European Union or the European Research Council Executive Agency. Neither the European Union nor the granting authority can be held responsible for them. We acknowledge the foundational contributions of former colleague Deenah Morton (née Osman), David C. Robinson for *P. sativum* for hydroponic analyses, and Matthew James Smith for checking the TEA. Diagrams in Figs. 1, 3, 4 and Supplementary Fig. 2 prepared with biorender.com.

## Author contributions

B.M.E., S.G.J., N.P.M., J.S., S.S., D.A.L., L.R.M., T.H., S.E.C., J.T., and A.W. designed and conducted experimentation. B.M.E., N.P.M., J.S., L.L.B.D. analyzed and presented data. B.M.E., N.P.M., J.S., L.L.B.D., N.R., K.A.F., C.H.E., J.C., J.F., M.W. and A.N.D. interpreted data and wrote the manuscript. B.M.E., K.A.F., C.H.E., J.C., J.F., M.W., and A.N.D. conceived the project, obtained funding, and supervised the project.

## Competing interests

Authors J.T., A.W., L.R.M. and J.F. are employees of LettUs Grow, which designs and manufactures indoor farming technology. None of the authors in academic institutions have commercial or personal financial interests in LettUs Grow.
