## [Transparent Peer Review file · Communications Biology]

Addressing Vitamin B12 deficiency through aeroponic fortification of a salad crop (*Pisum sativum*)

Corresponding Author: Professor Antony Dodd

This manuscript has been previously reviewed at another journal. This document only contains information relating to versions considered at Communications Biology.

Version 0:

Reviewer comments:

Reviewer #1

(Remarks to the Author)

The authors have thoroughly addressed the comments from the first review and added content that strengthened the manuscript, making it more directly relevant to their claim of developing a strategy for B12 fortification. The improvements of the manuscript are significant and highlight the novelty of the work. Therefore, I recommend publication.

I have some minor points to improve clarity, accuracy, and readability of the manuscript

Lines 57 & 59 - Change statement from just bacteria to prokaryotes or add archaea.

Figure 2B - The authors could emphasize (e.g. with a note in the legend) the difference in Y axis scale to highlight the change in mass needed to hit RDA.

Fig 3 A-C The cartoon model could be shown without repeating the smaller diagrams

Statements in lines 281, 284 are redundant.

Figure S2 - The authors should explicitly state the currency.

Reviewer #4

(Remarks to the Author)

I feel that the authors have addressed my comments and concerns sufficiently, I consider the manuscript appropriate for publication in Communications Biology, and I endorse its acceptance.

Reviewer #1 (Remarks to the Author):

This manuscript details the establishment of an aeroponics approach to fortify pea shoots with Cyanocobalamin (Vitamin B12). The researchers also validate the stability of the fortified cobalamin in cold-stored produce for 4 weeks and assess the accessibility of the cobalamin to the human digestive system using an in vitro tool. While the experiments are solid and the results are convincing overall, I do not believe the potential impact of this study merits publication in Nature Foods. Similar methods have already been published demonstrating mechanisms to deliver cobalamin into plants via the roots (Lawrence AD, Nemoto-Smith E, Deery E, Baker JA, Schroeder S, Brown DG, Tullet JMA, Howard MJ, Brown IR, Smith AG, Boshoff HI, Barry CE, Warren MJ. 2018. Construction of Fluorescent Analogs to Follow the Uptake and Distribution of Cobalamin (Vitamin B12) in Bacteria, Worms, and Plants. *Cell Chemical Biology* 25:941-951.e6.) and soaked seeds (Sato K, Kudo Y, Muramatsu K. 2004. Incorporation of a high level of vitamin B12 into a vegetable, kaiware daikon (Japanese radish sprout), by the absorption from its seeds. *Biochimica et Biophysica Acta (BBA) - General Subjects* 1672:135–137.). The authors improve upon the efficiency of this process, but it is still notably inefficient and potentially impractical at scale. I believe the article is more suitable for a more specialized journal.

Response: We thank the reviewer for their time. This has been extremely helpful in highlighting that we did not explain the novelty of our study clearly enough in the previous submission. The previous papers mentioned by the reviewer are helpful, with a key point of differentiation of our work being that we establish and validate an approach for incorporation of the RDA of B12 into salad that can be deployed commercially at scale. We have also added a techno-economic assessment, which demonstrates commercial viability.

Lawrence et al. (2018) includes a single image of fluorescent-tagged B12 in plant cells after supplying this molecule to the roots. Sato et al. is a brief report of a seed-soaking B12 fortification method, similar to a patent that we cite. This produces a reasonable level of B12 within the shoots, but it is unclear whether it is stable or the method is scalable.

Our approach differs from existing studies because we demonstrate and validate application within a scalable commercial growing system. We revised our manuscript to emphasize the points of novelty, and how we build on previous work. We revised the Introduction to explain that it is known that plants can absorb B12 from their growth media. We clarify that despite this, it is unknown whether this is useful or has potential for commercial deployment. The Introduction now explains, “We reasoned that there is potential to fortify salad products with B12 during their cultivation, since B12 can be absorbed by certain plants when present in the growth medium...” (lines 87-89) and “However, it is unclear whether approaches that involve supplementing plant growth media with B12 leads to a salad product that contains the RDA of B12, whether it can be absorbed by human digestion, and whether such approaches are economically viable or scalable.” (lines 91-94). We also refer to one of the prior studies in the Results (“Because B12 can be absorbed by seeds that are soaked in B12 prior to germination [27] and can accumulate within the cellular vacuoles of some plant species [21], we reasoned that aeroponically cultivated pea shoots might absorb B12 through their roots and accumulate B12 within harvestable plant shoots.” (lines 118-121). Finally, in response to another reviewer’s comment, we prepared a table comparing approaches in previous studies for B12 fortification on various growing platforms (hydroponics and seed soaking; Table S1). We also included a new small-scale comparison with hydroponics that we conducted. We think this makes clearer what was known versus what is new in our study, and allows us to show how our approach- going all the way from growing trials to scale economics- is unique.

The reviewer commented that the approach may be impractical / inefficient at scale. The reviewer might wish to learn that our approach has been scaled up for commercialization by a major UK supermarket. Based on the reviewer’s comment about efficiency, we conducted

a techno-economic assessment (TEA) of the approach and added a new section to the Results that explain its outcomes (lines 277-321), along with relevant Methods and Discussion sections. We appreciate the reviewer's encouragement to explain this better.

I have one major concern about the data, namely the inconsistent quantity of cobalamin detected upon fortification. Figure 1 E & F [*now Fig 2C, D*] show the results of two experiments supplementing pea shoots with cobalamin at different concentrations. However, the results are not replicated consistently. The authors should acknowledge this variation and its source, or revisit their methods and repeat the experiment.

Response: We agree the data are variable. The variability arises because the experiments were conducted in a commercial growing environment rather than under tightly controlled laboratory growth conditions. We deliberately chose this approach because our goal was to evaluate fortification under conditions that reflect the realities of a commercial growing environment, where gradients in many environmental variables such as light, temperature, humidity, planting density, and edge effects are much greater than in laboratory growth chambers. This focus on real growing environments is a strength that sets our work aside from previous studies. We believe this establishes a framework for agronomists and producers to establish and validate reliable fortification strategies within their own environments.

We recognise that this variability required explicit consideration within our study. To resolve this, we revised the paper in several ways, including with new data. (1) We revised the Abstract, Introduction, Results and Discussion to explain that a commercial growing environment was used, the rationale for doing this, and why we considered this to be a strength (Abstract (lines 35-36), Introduction (line 94), Results (lines 110-111), Discussion (324-325)). (2) We revised the Results to explain that the B12 levels varied between experimental repeats, yet delivered the RDA of vitamin B12 ("Although the overall B12 accumulation varied between experimental repeats (Fig 1C, D), a consistent feature was that B12 supplementation at 10 μ M and 20 μ M cyanocobalamin led to sufficient..." (lines 155-157)). (3) We conducted a PCA on variation in B12 accumulation within leaves, which suggests that variation in B12 accumulation associates with physical location in farm, and nutrient solution concentration (see below). We decided to not include this in the manuscript because we feel it is a distraction from the finding that irrespective of location in the farm, the pea shoots always accumulated in excess of the RDA. (4) We conducted and added a series of 5 independent growing trials to examine further the consistency of B12 accumulation between repeats. This identified significant variation in B12 accumulation, yet the plants always exceeded the RDA of B12 within a single portion of salad (15 g). This is described in a new Results section (lines 163-174), and new figure panel (Fig. 2E).

We thank the reviewer raising this, as it prompted us to both strengthen our explanation of the rationale behind our experimental strategy and clarify how we see this work as a roadmap for future fortification method development.

PCA: Environmental factors associated with variation in B₁₂ accumulation within pea shoot leaves. PCA analysis based on experiments in Figure 2D.

Other concerns that would benefit from the authors' attention include:

1. Generalization of the findings from one species of pea shoot to "salad crops". Either the findings should be limited to the species tested or another experiment should be performed that tests other phylogenetically diverse salad crops and include the results in the study.

Response: We agree this was misleading. We revised the text throughout, including the title and abstract, to clarify the species that was used. We preferred using a single species because it allowed us to (i) work at greater depth across multiple steps in the fortification process, and (ii) because pea shoots are of direct commercial relevance. If we worked with multiple species, it would dilute our effort and reduce the types of information we could obtain.

2. The x-axis labels ("Cyanocobalamin concentration in nutrient mix") in Figure 1 C-F [now Fig 2A-E] are difficult to see. Larger text or more space below the label would help.

Response: Thank you. This text should now be the same size as elsewhere in the figure.

3. Addressing the inconsistent shoot masses between the two methods of cobalamin supplementation, reported in Figure 1D [now Fig. 2B]. Why are the masses so different between the two methods? Are the different masses due to developmental stages between the treatments? Why is there more shoot mass with the lowest dose of B₁₂? How do these shoot masses compare to a control plant with no B₁₂?

Response: Unfortunately, the previous presentation of Fig. 2B produced a misunderstanding of the graph's content. This is not a measure of shoot mass. It shows the amount of pea shoots that must be harvested / eaten to obtain plant tissue that contains the

RDA of Vitamin B12. The problem was that Fig. 2B was not labelled clearly, and the associated text did not explain it properly. To resolve this we (1) rewrote the y axes on Fig. 2B to, "Pea shoot mass (g) required to reach B12 RDA (2.4ug)," (2) rewrote the associated text for the figure legend ("Comparison of mass of pea shoots that must be harvested in order to obtain plant tissue that contains the USA RDA of B12 (2.4 µg), when pea shoots were fortified using two different methods." Lines 685-687), (3) revised the relevant Results text ("Using these data, we calculated the quantity of pea shoots that need to be harvested to obtain plant tissue that contains..." (lines 140-141)). We apologise for the misunderstanding and thank the reviewer for noticing this.

4. Acknowledgement of possible microbial contribution of cobalamin to the fortification of plants, especially in legumes. *Pisum sativum*, the crop used in this study, is a legume which often form associations with root-nodule forming bacteria that are often B12-producers. It is known and has been shown that legumes such as soybeans (*Glycine max*) can obtain B12 from their microbial symbionts via their roots. (Mozafar, A., Oertli, J.J. Uptake of a microbially-produced vitamin (B12) by soybean roots. *Plant Soil* 139, 23–30 (1992). <https://doi.org/10.1007/BF00012838>)

Response: We agree with this. Such approaches could overcome the high cost of purified cyanocobalamin. In our experiments, the crop did not form root nodules (e.g. Fig. 1B). Nod factor (required for nodulation) would be absent from the nutrient mix in the relatively clean growing environment, and the crop was grown for only a short time before harvest such that nodules would not have time to form. We reviewed the literature, including the study recommended by the reviewer, and added an additional paragraph to the Discussion to consider this fortification strategy (lines 359-373). We thank the reviewer for recommending this interesting point.

5. Acknowledgement of the conflict of interest of authors associated with LettusGrow

Response: We agree. This was recorded on the manuscript submission system but not visible to the reviewer. Therefore, we revised our manuscript to include a clear conflict of interest statement.

Reviewer #3 (Remarks to the Author):

The authors have done very thorough and innovative work. These results are novel and the paper is well written. There are some improvements that could be made in the methodology and discussion sections to clarify and further provide context to the results. Please see the uploaded pdf for further details.

Response: We thank the reviewer for their encouragement and helpful feedback. It is tricky to write a response to a marked-up PDF; we attempted this below.

Add more details of existing strategies for biofortification of green vegetables.

Response: We agree with the need for better comparison with the literature. We took multiple steps to address this. (i) We added a table comparing literature for previous studies of B12 fortification using approaches such as hydroponics (Table S1). We refer to this in both the Introduction and Discussion. This helps provide a balanced demonstration of the merits and demerits of the approach we describe. (ii) We cite additional literature in the Introduction when describing the scientific history of the approach that we adopted. (iii) In the Discussion, we use existing literature to consider alternative approaches for B12 fortification, such as B12-overproducing bacteria and fermentation, and potential effects of nodulation and rhizobia on B12 content. We discuss the possibility that B12 uptake by plants might not be an active process, based on studies in soybean (lines 402-405). We also use previous

studies to emphasize a key distinguishing feature of our study, namely our multidisciplinary approach (plant sciences, engineering, economics, simulated human biology) to establish a roadmap for the process of developing and validating a fortification strategy. We feel that this last point is really important, but was not explained at all well in our previous submission.

Divide Methods up / unclear which cultivation method used for which experiment

Response: Thank you for noticing this. We reviewed it and found inconsistencies in certain information (e.g. growth duration of plants), and a lack of clarity about which methods mapped to which dataset. We revised the Methods to improve this, providing explicit mapping between each experimental approach and each figure. Furthermore, we had omitted description of plant growth for senescence and INFOGEST simulated digestion experiments, so revised the Methods to add this information and confirmed that brief details are also included within the figure legends. We hope this is now clearer.

Not all plant developmental stages used were microgreens, consider different wording.

Response: Thank you, we revised this throughout the manuscript.

What do you mean by 'more sustainable diets'?

Response: We revised this for clarification ("nutritional insufficiency risk as people adopt more sustainable diets that exclude meat", line 45). We also revised the abstract ("Plants do not produce Vitamin B12, presenting a risk of nutrient insufficiency for people that do not consume animal-derived foods." Lines 28-29). Thanks for asking for more information about this.

Which intrinsic factor?

Response: The protein is, indeed, called intrinsic factor. Since some readers will be unfamiliar with this aspect of human digestion, we extended the explanation ("...might prevent optimal B12 absorption because intrinsic factor (a glycoprotein that complexes with B12 and is essential its uptake)..." lines 73-74).

Discuss further the distribution of B12 within the plants.

Response: We added some further explanation but are wary of over-reaching. We added speculative discussion that we hope will prompt further work on the topic ("Accumulation of B12 in the leaves could occur through the action of the transpiration stream, given that cyanocobalamin is water soluble. If this is the case, it might be possible to increase B12 accumulation in the leaves through environmental treatments that maximise stomatal aperture during the B12 treatment period, or that take advantage of the times of day of maximum stomatal aperture. The accumulation of B12 in the roots could occur due to a restriction of uptake by the Casparian strip, but understanding this will require detailed knowledge of the uptake pathway." Lines 386-392). We also described some work in soybean that suggests uptake is not an active process (lines 401-404).

Discuss differences in B12 accumulation across plant developmental stages.

Response: We agree this merits brief discussion, being careful about over-speculation. We added, "Greater B12 accumulation after the cotyledon stage (Fig. 3) could simply reflect a greater tissue volume to store B12 in larger plants, or developmental stage-specific changes in vascular configuration or root anatomy." Lines 392-394.

Change "broken line" to "dashed line"

Response: Revised throughout.

Move description explanation of Fv/Fm to the Methods

Response: We would like to ensure our study is accessible to readers that are not plant biologists, so feel it is important to provide some explanation in the Results that is meaningful for those outside the field. Therefore, we retained some description in the Results, but also expanded on the definitions and detail of Fv/Fm in the Methods (lines 561-566).

Clarify what is being compared to what in the senescence time-series.

Response: Thank you, we revised this to, “During dark storage of the detached leaf tissue, relative to the start of the time series, electrolyte leakage increased...” (lines 223-225). We also revised the Figure 4 legend to clarify the statistical comparisons that were made in each figure panel (lines 705-735).

“In future” -> “In the future”

Response: Corrected (line 398).

More information about growth conditions should be provided

Response: From this comment, it is difficult to know which details are needed. We believe that we provided most of the standard information that is needed. We provided detailed information about the light, temperature, humidity, and nutrient supply conditions. However, in places, the mapping between certain experiments and the Methods section was not completely clear. Therefore, we revised the Methods and checked the figure legends to improve the detail. We also used other, specific, queries from the reviewer about plant growth methods to make further revisions to the Methods.

What developmental stage were plants at, at time of harvest?

Response: Thank you. We added this detail (“Pea shoots were cultivated in the aeroponic system for 8 days to the 3-4 node stage,” lines 462-463).

Add geographical location of equipment / reagent suppliers.

Response: Thank you. We checked the Methods and added these details where needed.

“Light tight” -> “Light excluding”

Response: We corrected this (line 480, 562).

Specify how many replicate growing trays were used

Response: Apologies for this oversight. The information has now been added to the Methods (“A minimum of 4 replicate trays were sampled per treatment (specified in figure legends).” Lines 503-504).

“jute matt” -> “jute mat”

Response: Corrected (line 495-496)

Root or root mat?

Response: Apologies for the confusion. It should have simply said roots (corrected to, “excess water was carefully squeezed from the roots before weighing (this was necessary because aeroponic irrigation leads to considerable amounts of water within the roots)” lines 497-499).

pH7.8 -> pH 7.8

Response: Corrected (line 511).

Most electrolyte leakage assays use autoclaving to disrupt all tissue. Please cite literature in support of the methods used

Response: To study electrolyte leakage, freezing can be used to release all electrolytes, as can autoclaving or boiling. We revised our manuscript to cite three papers that use this approach (“To release all electrolytes, leaf discs were transferred to a clean tube, and frozen to -70 °C for 1 h and then allowed to thaw gradually [53-55].” Lines 540-542). We also made a revision to specifically refer to a paper that explains the protocol used (“Electrolyte leakage due to cell degradation during senescence was measured using an established method [53]”. Lines 534-535). We thank the reviewer for requesting this clarification.

Chlorophyll fluorescence: how were plants dark adapted and were multiple leaves measured?

Response: We used chlorophyll fluorescence imaging rather than a leaf clip/fibre optic instrument, because the plants were small and fragile. This allowed a combined set of measurements from all leaves. To clarify this, we revised the Methods, “Because plants were small and fragile, we used chlorophyll fluorescence imaging rather than an approach using a leaf clip and fibre optic.” (lines 556-557), “Leaves were dark adapted for 20 minutes prior to measurement by placing the plant material in a light-excluding container” (lines 561-562) and “...with the mean fluorescence signal obtained across the entire leaf surface and all leaves on the plant measured.” (lines 567-569).

Split Fig. 1 into two separate figures.

Response: This is a great idea, as it has two conceptual parts. Therefore, we split the figure and associated legends into two, and renumbered throughout.

Clarify on the figure that Fig. 1E and F are two independent studies.

Response: This is correct, they are completely independent. Therefore, we revised Fig. 2C and D to indicate on the graphs directly that they are separate experiments. It is also now explained in the figure legend.

Response: On reservoir volume, we revised the Methods to add this (“The cultivation bed volume was 20 L and reservoir volume was 1000 L.” Line 471)

Reviewer #4 (Remarks to the Author):

The authors report on a method for fortification of a microgreen pea with vitamin B-12, using cyanocobalamin in aeroponically-supplied nutrient solution during horticulture cultivation. The results demonstrate the potential in reaching dietary sufficiency of this compound, within a reasonable serving. The experiments conducted adequately support the claims. The manuscript can benefit from putting the result in a broader perspective, for which I added

some additional questions below. The obtained knowledge provided a great added value to the current knowledge of B12 fortification of food crops. Overall, the manuscript is in a state in which it can be published, given that the points raised below are addressed.

Response: We appreciate the reviewer's option that the manuscript is close to publication standard, and their suggestions to broaden the perspective.

Abstract

Line 25: Plant-based foods lacking B12 does not consider plant-based fermented foods (though these are often trace amounts, for clarity this could be simply mentioning plants as a food source, which lack B12).

Response: Thank you. We made several revisions to address this. We revised the abstract to begin, "Plant do not produce Vitamin B12" and the start of the introduction to state, "only essential vitamin that is absent from plants" (line 44). We also revised the Discussion to mention this, noting from the literature that it seems unclear whether fermented products can contribute to serum B12 levels (e.g. <https://doi.org/10.1002/jsf2.137>) ("Another way that plant-based foods might be supplemented with B12 is through fermentation, although it is unclear whether the B12 present in products such as natto contributes to serum B12 levels [32]." Lines 371-373).

Introduction

Line 40: It is not clear how the limited number of enzymatic steps in which B12 serves as a cofactor relate to it only being needed in minute quantities. If these reactions only require B12 in very low concentrations, that would be another explanation. The lower RDA of B12 compared to other (B)-vitamins, could also be a perfect point to discuss its stability (half-life) in humans.

Response: Thank you, we realise our wording and logic was unclear. Therefore, we revised this part of the Introduction and refer to further literature. The small RDA is explained by the nutrient's exceptionally long half-life in humans and efficient intracellular recycling, rather than simply by the number of B12-dependent enzymes. To address this, we revised the Introduction to make this explicit and to cite the relevant literature. "Humans use the nutrient as a cofactor for just two enzymes, methionine synthase and methylmalonyl-CoA mutase [2]. The very long half-life of B12 means that the nutrient is required in only minute quantities daily [2-5], with the USA recommended daily allowance (RDA) for B12 being..." (lines 46-49).

Line 63: A bit of information seems to be missing here. Briefly address these reasons (a shift towards a plant-based diet being one?)

Response: We agree that this needed rewriting. We modified these sentences to, "Despite the presence of B12 in many dietary sources, it is important for those that are B12 deficient or insufficient to supplement their nutrition with B12." (lines 71-72).

Line 67: Here, a quick overview of how this fortification is happening (globally in cereal and dairy products) could be beneficial.

Response: This is a nice point. We revised the Introduction, and also discussed some additional literature, "An alternative approach to B12 tablets is fortified food products, which can be consumed as part of a nutritious diet that promotes optimal B12 absorption. This includes fortification of a variety of foods, such as breakfast cereals, milk, and wheat flour [2], with B12 fortification of flour occurring in many countries [16]. An alternative approach considered here is the fortification of fresh salad products." (lines 75-80).

Line 72: Interesting take on rising popularity of these food products. Is inflation considered in the statement of these numbers?

Response: We double checked this. The CAGR calculations are not inflation-adjusted. We noted this in the text (line 85). Thanks for spotting that.

Results

Line 111: To avoid confusion, here aeroponic growth could be mentioned again.

Response: Good point; revised (now line 120).

Discussion

Line 260: This is an interesting point, could this be extended to other vitamins or minerals?

Response: We revised the Discussion to elaborate a little, but are cautious about over-reaching. We explained that the approach could be extended, mentioning the requirement for optimization and validation, and the potential for combination with genetic approaches for biofortification (“It is possible that aeroponic cultivation could be used to deliver other vitamins or minerals into salads in a relatively straightforward manner, although it would require experimental optimization and validation for any combination of nutrient and crop. Furthermore, we reason that the method could be combined with other fortification methods, such genetic approaches to increase iron accumulation within plant tissue, to fortify salads in a combinatorial manner.” Lines 346-351).

We feel that the combination of approaches that we used (underpinning biology, economic models, scalability, validation) establishes a pathway for development and deployment of crop fortification, and now explain this point to the Discussion (“Focusing on multiple aspects of B12 fortification in a single species has enabled us providing a roadmap for this type of crop fortification in other species and with other nutrients, from the underpinning biology to an economic evaluation.” Lines 433-436). Thank you, this has added impact to our manuscript.

Line 296: the example of usage in isolated space environment raises the question whether simply using the chemical B12 compounds directly for human consumption would not be more efficient? Is it correct to relate to the benefit of these fortified microgreens as being perceived more ‘natural’ compared to simple direct supplementation?

Response: After considering the reviewer’s comment, we realize this was over-reach and removed it. We think it distracted from the focus of our article. Thanks for asking about this.

Line 307-308: This seems very promising. However, do correctly make the statement on an economically viable production at scale, some of questions related to costs and product premium would need to be addressed (see also further comments below).

Response: This is a welcome comment. In response to this and a question from another reviewer, we conducted an economic evaluation on the potential additional cost of our approach in practice, and at scale. We added a new section to the Results that explains the model outcomes (lines 277-321, Fig. S2). We revised the Abstract to explain that we conducted an economic evaluation (line 36). We also made revisions throughout to explain the approach is scalable and the advantages of having conducted the experiments in a real-world growing environment. The reviewer might be interested that our approach has been adopted by a UK supermarket, but we feel it is not appropriate to discuss that in a scientific paper.

Line 358-359: phrasing seems a bit weird here (“from”).

Response: Agreed; revised to, “roots were cut from underneath the jute mat” (line 497).

Comments and questions:

Is B12 stability in water a problem, both physiologically and economically (discussing the loss of the relatively expensive compound)?

Response: B12 stability is an issue under certain conditions. Cyanocobalamin degrades faster outside a pH range of ~ 4-6.5, and is UV light sensitive. The latter is important as UV light is sometimes used for nutrient solution sterilization in indoor farms. In our approach, the nutrient mix was excluded from the UV treatment for the duration of exposure of plants to cyanocobalamin, and the nutrient mix pH was controlled. We realise that we missed some details from the Methods. We revised the Methods to include, “Cyanocobalamin can degrade outside a pH range of 4-6.5 and in the presence of UV light [51] (one study indicates it has greatest stability at pH 6.3 [52]). Therefore, for the duration of the cyanocobalamin treatment of the plants, the pH was carefully controlled (pH 6.2, as above) and the nutrient mix was disconnected from the reservoir so that it was not exposed to a sterilizing UV treatment.” (lines 485-489).

To assess whether this is a realistic fortification method, what is the expected cost of deploying this large-scale, how much would the price premium of the fortified microgreens be?

Response: Great point. This led us to conduct a techno-economic assessment of the approach, which is described in a new Results section (lines 277-321). Our models estimate the *additional* cost of fortification, rather than the overall cost of indoor growing, as the latter depends a lot on region of the world, energy costs, etc. In short, it is economically viable.

How does this compare to simple supplementation, and growth of plants with overproducing bacteria?

Response: We find the strategy of using B12-overproducing bacteria very interesting. We do not know how the economics of this would work out: on the one hand, it removes the need to source cyanocobalamin, on the other hand it might present difficulties with food safety regulators and require modifications to the indoor farming equipment. We agree we should discuss this, and added a section to the Discussion on this topic (lines 351-358). We thank the reviewer for bringing this to our attention- it’s a nice discussion point and alternative.

Is it economically viable, is there a decent amount of loss in cyanocobalamin (how expensive is this compound)?

Response: This is an important question. In our techno-economic assessment, we make the point that cyanocobalamin is expensive, and modelled scenarios involving three different grades of cyanocobalamin. We observe that the cyanocobalamin persists within the nutrient mix after the fortification period- based on the colour of the nutrient mix- but do not have direct measures of this. Our economic models simulate both no nutrient mix reuse, and also nutrient mix reuse with cyanocobalamin loss. Both approaches seem economically viable. We describe these results within the economic evaluation section and associated figure (line 291-293, 304-307, Fig. S1). Thank you for raising this point.

How does cyanocobalamin compare to other chemical forms like adenosylcobalamin regarding bioavailability and bioactivity, stability).

Response: Adenosylcobalamin and methylcobalamin are the two coenzyme forms of vitamin B12 that are directly bioactive in humans. However, they are extremely unstable, particularly to light and heat, with adenosylcobalamin degrading in seconds. This instability makes them unsuitable for food fortification as they degrade before consumption.

Hydroxocobalamin is more stable than the coenzyme forms and is readily converted into the active cofactors, since it requires only one intracellular conversion step. However, it is more expensive, less widely available for food applications, and less stable than cyanocobalamin under typical storage conditions. It is also prioritised for clinical use (e.g. injectable B12 therapy for pernicious anaemia), which further limits its practicality for food fortification at scale.

By contrast, cyanocobalamin is stable during storage and handling, and relatively inexpensive, so is the most widely used commercial form. Although it requires at least two intracellular conversion steps into adenosylcobalamin and methylcobalamin, this occurs efficiently in humans without malabsorption disorders such as pernicious anaemia, meaning it is bioavailable and bioactive. Therefore, cyanocobalamin is considered the most practical form for food fortification because it balances stability with reliable biological activity. We have clarified this rationale in the manuscript (“We chose to use cyanocobalamin rather than other forms of the nutrient, because this is the most widely available form that is also bioactive for humans and has greater stability than adenosylcobalamin and methylcobalamin, which photodegrade in seconds [30].” Lines 115-118).

Is the requirement of chemical B12 a limiting factor in this process, i.e. how does it compare to methods overproducing B12 by cocultivation with B12-producing bacteria (<https://doi.org/10.1002/jsfa.14095>). Would the requirement make the fortification approach more difficult in rural regions around globally?

Response: Our approach is particularly appropriate in situations where salads are produced already using indoor farming, as it is a straightforward adaptation to existing technology. Indoor farms have energy and capital costs that might not be an option in some regions, so approaches such as B12-producing bacteria could be advantageous. We mention this in the Discussion, as it is an important consideration (“As the strategy that we describe uses indoor farming technology that has increased energy requirements and requires capital expenditure, it might not be suitable for B12 fortification in all regions globally, so approaches such as fortification using B12-producing bacteria [42] could offer a less resource-intensive alternative.” (lines 367-371).

Do pea seedlings or microgreens in general, cultivated in system which would allow incorporation of the described B12 supplementation method, enable reaching a great subset of the human population? This is a valid point of discussion, perhaps the target population, people shifting to plant-based diets, are more easily reached?

Response: Our data indicate that the B12 persists within pea shoot tissue for at least 4 weeks, which could (in theory) allow distribution almost anywhere globally. We do not want to over-reach but added to the Discussion consideration of this, “Given that the B12 persists within the plant tissue for at least 4 weeks (Fig. 4B), the material could- in theory- be distributed to most locations globally if the economic case existed.” (lines 331-333).

Reviewer #5 (Remarks to the Author):

The manuscript by Eldridge et al., "Addressing Vitamin B12 Deficiency through Aeroponic Fortification of Salad Crops," explores a promising approach to addressing vitamin B12 deficiency, particularly for individuals following a vegan diet. The authors propose aeroponic

fortification as an effective method to deliver the recommended daily allowance of vitamin B12 in a single serving of microgreen salad crops.

Although the concept is intriguing, the study lacks comparative data with other methodologies, making it difficult to determine the efficiency of the proposed method. The discussion section emphasizes the benefits of aeroponic cultivation in promoting root development and vitamin B12 uptake compared to hydroponics; however, these claims are not supported by the presented results. Additionally, the potential for reusing the B12 solution, as mentioned in the discussion, should be investigated to determine whether reuse increases the vitamin B12 content in seedlings. Finally, the study was conducted on a single microgreen variety, and to demonstrate the effectiveness of aeroponics for B12 biofortification in salad crops, it should be tested on at least one other microgreen variety.

Response: We thank the reviewer for these thoughtful comments, which helped us recognise the need to clearly articulate the scope and the novelty of our study. Our purpose was not only to demonstrate that aeroponic fortification can reliably deliver the RDA of vitamin B12 in a commercially relevant crop, but also provide a roadmap for evaluating and scaling fortification methods. To do this, we deliberately adopted an interdisciplinary approach where we validated fortification under commercial growing conditions, coupling this with a simulated bioaccessibility study widely used in food science, and testing nutrient stability post-harvest using validated approaches from plant science. We now also complement these biological studies with a simple techno-economic analysis, which quantifies fortification and cost identifies opportunities for optimisation.

Following the reviewer's advice, we strengthened the manuscript in several ways. We rewrote parts of the Introduction, Results, and Discussion to clarify our rationale and methods, added comparative discussion with other methodologies, incorporated new data and analyses, and highlighted how our results inform future work. This makes clear that our work is not only a proof-of-concept but also a foundation for developing, validating, and scaling other nutrient fortification strategies in commercially viable crops. We believe this integrated approach is a unique contribution is well suited to Nature Food.

For these reasons I suggest:

1. To conduct experiments comparing aeroponic fortification with other methods, such as hydroponics, to provide clearer insights into the relative efficiency of each technique for vitamin B12 biofortification.

Response: There are existing studies adopting various hydroponic approaches for B12 fortification of salad crops. We decided to assemble and summarise this literature in a supplemental table within our study (Table S1). We also conducted a small-scale experimental comparison with hydroponics, included these data in the table, and describe the outcomes (lines 335-337). It is quite informative to include these comparisons with the work of others- we thank the reviewer for recommending this.

We agree that our claim that the method we describe is more "efficient" than other approaches was inappropriate. Therefore, we revised the manuscript throughout to remove claims about relative efficiency. A claim of efficiency also ultimately depends on local economics, so it is not straightforward to make such a claim.

2. To test the aeroponic fortification method on other salad crop species to evaluate its effectiveness across different plants and ensure that the findings are not specific to a single variety.

Response: We thank the reviewer for this comment. We focused on a single species to gain depth rather than breadth of insight. This strategy allowed us to evaluate all stages of the

fortification process, from cultivation under commercial conditions through to nutrient bioaccessibility and a techno-economic analysis, thus providing a more comprehensive and impactful validation than would have been possible by spreading resources across multiple species.

Based on existing literature, we believe the approach could be adapted to other crops, but each would require proper validation. To clarify this, we revised the Discussion to note: “Previous studies reported that hydroponics can be used to fortify certain additional species with B12 (Table S1), suggesting this could be adapted to aeroponics. However, approaches with additional species require work to determine whether the B12 is bioaccessible during digestion, is accumulated in forms that are active in humans, and to evaluate commercially viability” (lines 335-339). We also revised the end of the Discussion to explain why our chosen strategy is a strength: “Focusing on multiple aspects of B12 fortification in a single species has enabled us providing a roadmap for this type of crop fortification in other species and with other nutrients, from the underpinning biology to an economic evaluation” (lines 433-436).

3. To Investigate the effects of reusing the B12 solution on the vitamin B12 content in seedlings. This would help determine whether reusing the solution is a viable and efficient approach for increasing B12 uptake.

Response: We thank the reviewer for this important suggestion. We agree that reuse of the B12 solution is a promising strategy. Experimentally, we are unable to test this directly because our horticulture platform uses UV sterilisation of nutrient solutions, and UV causes rapid degradation of cyanocobalamin. Nevertheless, we recognise the relevance of this approach and incorporated it into our new techno-economic analysis (lines 277-321). Specifically, we modelled the potential cost savings of recapturing and reusing the fortified nutrient solution across multiple growing cycles, while applying a loss function to account for degradation. This analysis identifies both the potential commercial benefits of reuse and the technical challenges that need to be solved, such as avoiding UV degradation during sterilisation.

We also revised the Discussion to reflect this, noting: “Growers might wish to recapture the B12 solution and use it again after root exposure to B12. One challenge with recapture and reuse that will need to be overcome is to avoid cyanocobalamin degradation by the UV light sources used to sterilize the nutrient mix in indoor growing environments” (lines 381-384).

4. To Explore the interactions between vitamin B12 and other nutrients in the aeroponic system to understand how different nutrients might affect the uptake and accumulation of vitamin B12 in the plants.

Response: We thank the reviewer for this suggestion. While it would be impractical to systematically test all possible nutrient-B12 interactions, we considered this issue in two ways. First, in small-scale preliminary experiments conducted before the results presented in this manuscript, we used a different horticultural nutrient formula with altered nutrient ratios. Even under these conditions, our fortification strategy consistently enabled a portion of pea shoots to meet the RDA for B12. These results are not included in the manuscript because of the size of the study, but strengthen our confidence that the method is robust to variation in nutrient formulation.

Second, during the study we collected environmental and nutrient-related variables, and performed an exploratory principal component analysis (PCA) to investigate factors associated with variation in B12 accumulation. This suggested that nutrient solution electrical conductivity, the physical location of the growing stack and bed (i.e., in-farm positional effects), and growing bed temperature were the strongest correlates of variability. As these

results are preliminary and not central to our main findings, we have chosen to not include them in the manuscript.

We think this topic should be mentioned and have revised the Discussion to include the observation, “It will also be informative to understand how different horticultural nutrient formulations affect B12 uptake,” (lines 401-402), as a way to guide future work.

Editorial comments

Please add missing units in Tables S1 and S2.

Response: Our apologies, we should have spotted this. Units have now been added.

In line 72, please change “essential its uptake” to “essential for its uptake”.

Response: We have corrected this.

Style changes

We shortened the abstract to meet the journal’s requirements.

We added a new supplementary figure that shows all individual replicates for the line graphs in Figure 4. In this case, each treatment is split onto a separate graph. We tried showing all treatments on the line graphs in Figure 4 and it was uninterpretable, so we reasoned better to include a new figure that shows this.

Reviewer 1

The authors have thoroughly addressed the comments from the first review and added content that strengthened the manuscript, making it more directly relevant to their claim of developing a strategy for B12 fortification. The improvements of the manuscript are significant and highlight the novelty of the work. Therefore, I recommend publication.

Response: We welcome this feedback.

I have some minor points to improve clarity, accuracy, and readability of the manuscript

Lines 57 & 59 - Change statement from just bacteria to prokaryotes or add archaea.

Response: Thank you- this has been corrected.

Figure 2B - The authors could emphasize (e.g. with a note in the legend) the difference in Y axis scale to highlight the change in mass needed to hit RDA.

Response: This is a great suggestion. We added a sentence of emphasis to the figure legend (“Note the differing y axis scales on the graphs for the two methods.”).

Fig 3 A-C The cartoon model could be shown without repeating the smaller diagrams

Response: With great respect, we do not agree. As this is a multidisciplinary study within a multidisciplinary journal, we wished to make the article accessible to those outside the field of plant sciences. Therefore, we have sought to produce figures that are easier for the broad readership of Communications Biology.

Statements in lines 281, 284 are redundant.

Response: Thank you, good point. We addressed this redundancy by removing the final sentence of this paragraph.

Figure S2 - The authors should explicitly state the currency.

Response: Thanks. We revised the labelling of the y axes the graphs in Fig. S2 to make this clearer.